# Cycle Consistency Driven Object Discovery

**Aniket Didolkar [1], Anirudh Goyal [2], Yoshua Bengio [1]**

## Abstract

Developing deep learning models that effectively learn object-centric representations, akin to human cognition, remains a challenging task. Existing approaches facilitate object discovery by representing objects as fixed-size vectors, called "slots" or "object files". While these approaches have shown promise in certain scenarios, they still exhibit certain limitations. First, they rely on architectural priors which can be unreliable and usually require meticulous engineering to identify the correct objects. Second, there has been a notable gap in investigating the practical utility of these representations in downstream tasks. To address the first limitation, we introduce a method that explicitly optimizes the constraint that each object in a scene should be associated with a distinct slot. We formalize this constraint by introducing consistency objectives which are cyclic in nature. By integrating these consistency objectives into various existing slot-based object-centric methods, we showcase substantial improvements in object-discovery performance. These enhancements consistently hold true across both synthetic and real-world scenes, underscoring the effectiveness and adaptability of the proposed approach. To tackle the second limitation, we apply the learned object-centric representations from the proposed method to two downstream reinforcement learning tasks, demonstrating considerable performance enhancements compared to conventional slot-based and monolithic representation learning methods. Our results suggest that the proposed approach not only improves object discovery, but also provides richer features for downstream tasks.

## 1 Introduction

Having object-based reasoning capabilities is important for solving many real-world problems. The world around us is complex, diverse, and full of distinct objects. Human beings possess the natural ability to parse and reason about these objects in their environment. Frequently, changing or manipulating certain aspects of the world requires interacting with a single object or a combination of objects. For instance, driving a car necessitates maneuvering a single object (the car) while avoiding collisions with other objects or entities such as other cars, trees, and other obstacles. Developing object-based reasoning capabilities is therefore crucial for improving the ability of deep learning models to understand and solve problems in the real world.

Unsupervised discovery of objects from a scene is a challenging problem, as the notion of what an object refers to may be hard to parse without any extra context. Many existing approaches (Greff et al., 2017; 2019; Burgess et al., 2019; Goyal et al., 2019; Locatello et al., 2020; Goyal et al., 2020; 2021b; Ke et al., 2021; Goyal et al., 2021a; Singh et al., 2022) learn a set of slots to represent objects, where each slot is a fixed size vector. Most of these approaches use a reconstruction loss coupled with certain architectural biases that depend on visual cues to learn to segment objects into slots. There has been work that uses other auxiliary cues for supervision such as optical flow (Kipf et al., 2021) and depth prediction (Elsayed et al., 2022). However, architectural priors may not be always reliable and hence may not scale to real-world data while relying on auxiliary information such as optical flow and motion is not feasible since many datasets and scenes do not come with this information. To address these limitations, we augment existing slot-based methods with two auxiliary objectives called *cycle consistency* objectives.

[01] Mila, University of Montreal, [2] Google DeepMind
Corresponding authors: `adidolkar123@gmail.com`

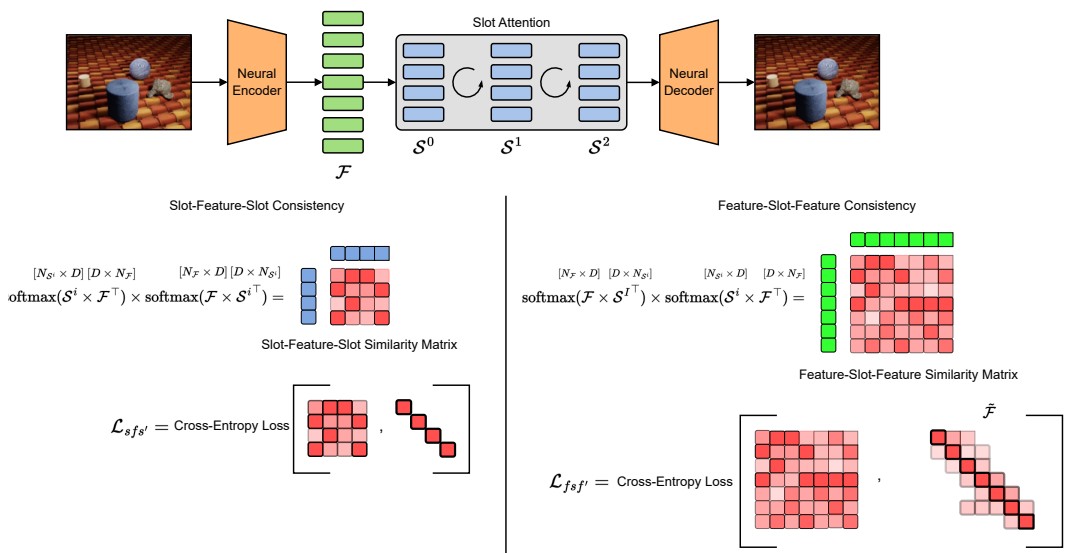

Figure 1: **Cycle Consistency Objectives**. Here we show the general architecture of the model. The proposed approached augments two additional losses to existing object-centric methods (in this case, slot attention). These losses encourage object discovery directly in the latent space. We refer to these losses as SLOT-FEATURE-SLOT Consistency loss and FEATURE-SLOT-FEATURE consistency loss. The SLOT-FEATURE-SLOT consistency loss is calculated as the cross-entropy between the slot-feature-slot similarity matrix and the identity matrix. The FEATURE-SLOT-FEATURE consistency loss is computed as the cross-entropy between the feature-slot-feature similarity matrix and a custom matrix $\tilde{\mathcal{F}}$, the details of which we provide in Section 2.

The proposed cycle consistency objectives operate directly on the latent representations of the slots and visual features (obtained from the neural encoder as shown in Figure 1). These objectives augment the architectural priors used in slot attention style models with an extra layer of reliability by explicitly enforcing coherence and consistency between the representations of the visual features and the learned slots. To apply the objectives, we consider the visual features and slots as nodes in a directed graph. The problem of object-discovery can be then formulated as that of adding the right edges into the graph such that - (1) the outgoing edges from a set of features belonging to the same object should go into the same slot, and, (2) the outgoing edges from each slot should go into a distinct subset of features. Both these constraints are formulated into two cycle consistency objectives called SLOT-FEATURE-SLOT consistency loss and FEATURE-SLOT-FEATURE consistency loss. Further details regarding these objectives are elaborated in Section 2.

The proposed objectives are simple and can be integrated into any existing slot-based object-discovery method. We apply them to two object-discovery tasks and two downstream RL tasks. We find that augmenting slot-based methods with the proposed objectives improves object-discovery performance and exhibits stronger generalization to unseen scenarios. The learned slot-based representations also demonstrate strong transferrability to downstreal RL tasks as compared to various baselines.

## 2 PROPOSED METHOD

In this section, we present the details of the proposed cycle consistency objectives and the underlying intuition behind them. The proposed method is designed to operate on a set of $N$ slots $\mathcal{S} = \{s_0, s_1, \ldots, s_N\}$ and a set of $M$ features $\mathcal{F} = \{f_0, f_1, \ldots, f_M\}$ which can be obtained using any suitable backbone such as a convolutional encoder. Similarly, the slots $\mathcal{S}$ can be obtained using any suitable object discovery or slot extractor method such as slot attention as shown in Figure 1:

$$\mathcal{S} = \text{slot\_extractor}(\mathcal{F}) \tag{1}$$

**Preliminary Setup** We denote the directed graph between the nodes and the features as $\mathcal{G}$. To find the correct edges to add to this graph we first score all possible edges. Next, we optimize the scores to satisfy the following two conditions -

- A set of features belonging to the same object must have outgoing edges to the same slot.

- The outgoing edges from each slot should go to a distinct set of features.

The score for an edge between a slot and a feature is computed by taking the dot product of their respective features. For example, an outgoing edge from feature $f_i$ to slot $s_j$ is scored as -

$$\phi(f_i, s_j) = \phi(f_i \to s_j) = \frac{f_i \cdot s_j}{\tau_1} \tag{2}$$

Here, $\tau_1$ is the temperature which is generally set to 0.1 in our experiments. We convert these scores into probabilities by normalizing across all possible target nodes.

$$p(f_i \to s_j) = \frac{\exp(\phi(f_i \to s_j))}{\sum_{k=0}^{k=N-1} \exp(\phi(f_i \to s_k))} \tag{3}$$

By computing these probabilities for all possible feature-slot pairs, we obtain a feature-slot similarity matrix $A_{f \to s} \in \mathbb{R}^{M \times N}$. Similarly, we compute a slot-feature similarity matrix $A_{s \to f} \in \mathbb{R}^{N \times M}$. Note that, the scores in $A_{s \to f}$ are normalized across all features.

We want both the similarity matrices to conform to the above two conditions. For example, each row in $A_{s \to f}$, which represents a particular slot $s_i$, should assign highest probability to the features corresponding to that slot. Similarly, each row in $A_{f \to s}$ should assign highest probability to the slots that the feature corresponds to.

However, we cannot directly optimize these similarity matrices to satisfy the above conditions as we do not have the ground truth slot to feature assignments. Instead, we consider cyclic paths consisting of two edges - (1) SLOT-FEATURE-SLOT paths - Paths with an edge from a slot to a feature and another edge from a feature to a slot; (2) FEATURE-SLOT-FEATURE paths - Paths with an edge from a feature to a slot and another edge from a slot to a feature.

**SLOT-FEATURE-SLOT Consistency Loss** We want that for an outgoing edge from slot $s_i$ to feature $f_k$, the outgoing edge from feature $f_k$ should cycle back to slot $s_i$. To develop an intuition about this, consider a case where perfect object factorization has been achieved where each object is represented by a distinct slot. In such a case, for an edge going from a slot $s_i$ to a feature $f_k$, the outgoing edge from $f_k$ will always come back to $s_i$ because in the case of perfect factorization each feature belonging to a particular object will be represented by one slot only - the slot that represents that particular object.

To achieve this constraint, we first calculate the scores of all possible SLOT-FEATURE-SLOT paths i.e. paths with edges from a slot $s_i$ to a feature $f_k$ and another edge from $f_k$ to slot $s_j$ as follows - $\phi(s_i \to f_k \to s_j) = p(s_i \to f_k) \cdot p(f_k \to s_j)$. We convert this into probabilities by normalizing over all possible paths beginning from slot $s_i$ and ending in slot $s_j$ as follows -

$$A_{s \to f \to s}[i, j] = \frac{\exp(p(s_i \to f_k) \cdot p(f_k \to s_j))}{\sum_{k'=0}^{M-1} \sum_{j'=0}^{N-1} \exp(p(s_i \to f_{k'}) \cdot p(f_{k'} \to s_{j'}))} \tag{4}$$

$$A_{s \to f \to s} = \text{softmax}(A_{s \to f} A_{f \to s}, axis = 1) \tag{5}$$

We refer to $A_{s \to f \to s} \in \mathbb{R}^{N \times N}$ as the slot-feature-slot similarity matrix as shown in the Figure 1. Since we want every path originating at slot $s_i$ to return back to slot $s_i$, we want the probabilities along the diagonal of $A_{s \to f \to s}$ to be the highest hence we frame the SLOT-FEATURE-SLOT consistency loss as -

$$\mathcal{L}_{sfs'} = -\sum_{i=0}^{N-1} \log(A_{s \to f \to s}[i, i]) \tag{6}$$

**FEATURE-SLOT-FEATURE Consistency Loss** For a path originating at a feature $f_i$, belonging to an object $o$, going into a slot $s_k$. The outgoing edge from slot $s_k$ must go into any of the features $f_j$ that represent $o$. Note that in this case the outgoing path from slot $s_k$ may not go back to the originating feature $f_i$ since multiple features belonging to one object map to the same slot.

To achieve this constraint, we first calculate the feature-slot-feature similarity matrix which is shown in Figure 1 as follows -

$$A_{f \to s \to f} = \text{softmax}(A_{f \to s} A_{s \to f}, axis = 1) \tag{7}$$

Note that $A_{f \to s \to f} \in \mathbb{R}^{M \times M}$. In this case, we cannot optimize for the diagonals of $A_{f \to s \to f}$ to have the highest probabilities since SLOT-FEATURE-SLOT paths can have different source and target nodes. Therefore, we optimize it by computing the cross-entropy between $A_{f \to s \to f}$ and a custom matrix $\tilde{\mathcal{F}}$ -

$$\mathcal{L}_{fsf'} = - \sum_{i=0}^{M-1} \tilde{\mathcal{F}}[i,i] \log(A_{f \to s \to f}[i,i]) \tag{8}$$

Note that the above loss is **only** computed for the diagonal elements of $A_{f \to s \to f}$.

$\tilde{\mathcal{F}}$ is calculated as a function of the features $\mathcal{F}$ output by the encoder as indicated in Figure 1. First we calculate the pairwise feature similarity values using $\mathcal{F}$ and sparsify the feature similarity matrix based on a threshold $T$. Consider two features from $\mathcal{F}$ - $f_i$ and $f_j$. The similarity score between these features is calculated as $\delta_{i,j} = \frac{f_i \cdot f_j}{\tau_2}$. The threshold value $T$ is computed as $T = c \cdot (\max(\mathcal{F}) - \min(\mathcal{F})) + \min(\mathcal{F})$, where $c$ is a hyperparameter. In all our experiments, we set $c$ to 0.8 unless specified otherwise. Once we obtain the sparse feature similarity matrix (denoted as $\tilde{F}$ which is a $M \times M$ matrix), we normalize it across rows to convert the similarity scores into probabilities -

$$\tilde{\mathcal{F}} = \text{softmax}(\tilde{F}, axis = 1) \tag{9}$$

**Training Details** The proposed method can be applied on top of any slot-based object discovery method. To incorporate the proposed approach, we add the the cycle consistency objectives to the original loss of the method. For example, slot attention (Locatello et al., 2020) utilizes a pixel-based reconstruction loss. On adding the proposed objectives, the final loss becomes:

$$\mathcal{L}_{final} = \mathcal{L}_{recon} + \lambda_{sfs'}\mathcal{L}_{sfs'} + \lambda_{fsf'}\mathcal{L}_{fsf'} \tag{10}$$

To set the hyperparameters for our approach, we select $\lambda_{sfs'}$ and $\lambda_{fsf'}$ as 0.1 and 0.01, respectively, unless otherwise specified. We also employ an additional Exponential Moving Average (EMA) visual encoder. The EMA encoder is used for calculating $\tilde{\mathcal{F}}$. This practice aligns with several self-supervised learning studies (He et al., 2019; Chen & He, 2020; Grill et al., 2020) and ensures that the features used in computing $\tilde{\mathcal{F}}$ remain stable avoiding frequent changes caused by gradient updates in the visual encoder. Also, we detach the calculation of $\tilde{\mathcal{F}}$ from the gradient computation.

In methods involving multiple iterations of object discovery, such as slot attention, we apply the cycle consistency objectives to the slots obtained from each iteration of the method, unless explicitly stated otherwise.

We set $\tau_1$ to 0.1 and $\tau_2$ to 0.01 unless specified otherwise.

**Connections to k-means clustering** Locatello et al. (2020) mention that slot attention can be seen as a form soft k-means clustering with dot product as the similarity function and a learned update rule. Taking this view, the proposed cycle consistency objectives can be seen as a further enforcing function on the clustering that acts by maximizing the similarity between features that belong to a cluster and minimizing the similarity of features that belong to seperate clusters.

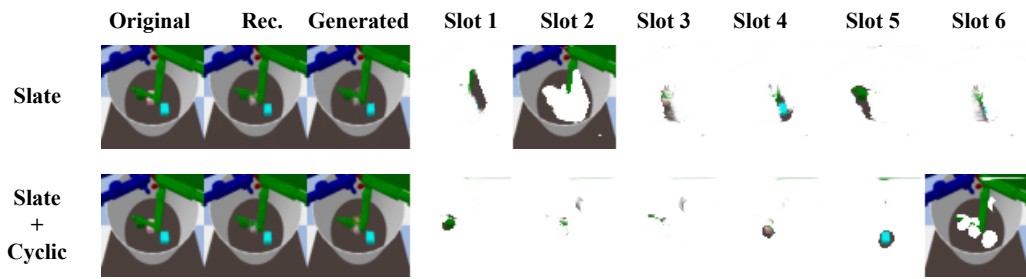

Figure 2: **Causal World Slot Masks** In this figure, we show the learned slot masks for both the pre-trained models. We can see that proposed approach captures obejcts more clearly than the baseline Slate model.

## 3 RELATED WORK

**Unsupervised Object Discovery** Our work addresses the problem of object discovery in visual scenes. While this challenge has been tackled using supervised, semi-supervised, and unsupervised methods, our approach falls into the category of unsupervised object discovery techniques (Greff et al., 2017; 2019; Burgess et al., 2019; Eslami et al., 2016; Lin et al., 2020; Goyal et al., 2020; Crawford & Pineau, 2019; Zoran et al., 2021; Locatello et al., 2020; Ke et al., 2021; Engelcke et al., 2019; Goyal et al., 2019; 2021b). These papers mainly consider only synthetic datasets. They rely on a set of vectors, known as *slots*, to represent objects, and use various architectural priors and objectives to group image features into slots that correspond to distinct objects. Many recent works (Singh et al., 2022; Jia et al., 2022; Seitzer et al., 2023; Choudhury et al., 2021; Wang et al., 2022; 2023) have proposed improvements that have managed to scale these slot-based methods to real-world datasets. While most of these datasets use pixel-wise reconstruction objectives, (Seitzer et al., 2023; Wang et al., 2023) are the only two works that utilize objectives in the latent space like the proposed approach. (Seitzer et al., 2023) introduce a model called Dinosaur which applies slot attention on the features output by a pretrained self-supervised encoder such as DINO (Caron et al., 2021) and use feature reconstruction as the training objective. (Wang et al., 2023) also apply slot attention to the features output by a pretrained encoder. They use an objective which is very similar to the cycle consistency objective proposed in this paper. The main difference between our work and theirs is that they mostly consider pretrained and frozen encoder backbones while we mainly consider encoders trained from scratch.

**Cycle Consistency** Cycle consistency is a concept in deep learning that enables the learning of a consistent mapping between two domains in cases where 1-to-1 data is not available. It relies on the property of transitivity to enforce structure in representations, and has been successfully used for learning good representations in images, videos, and language. Numerous studies (Wang et al., 2013; 2014; Wilson & Snavely, 2013; Zach et al., 2010; Zhou et al., 2015a; 2016; 2015b; Hoffman et al., 2018; Zhu et al., 2017) have employed cycle consistency in image-based tasks such as co-segmentation, domain transfer, and image matching. In these works, cycle consistency is typically used as an objective function that ensures the consistency of the mapping between the source and target domains, and the inverse mapping from the target domain back to the source domain. For example, in (Zhu et al., 2017), the source and target domains are images from distinct styles. Cycle consistency has also been utilized as a self-supervised representation learning technique in videos. Various studies (Dwibedi et al., 2019; Wang et al., 2019; Li et al., 2019; Lai & Xie, 2019; Jabri et al., 2020; Hadji et al., 2021) have used the cycle consistency objective to enforce temporal consistency in videos, ensuring that there is a path forward from frame $i$ to frame $k$, and the path backward from frame $k$ lands back on frame $i$. Our work differs from previous works in that we apply the cycle consistency objective for object discovery. Additionally, the cycle consistency objective is applied to the latent space consisting of the slots and features in this case while previous studies have primarily focused on applying the objective to the domains of language, images, or video.

## 4 EXPERIMENTS

In our experiments, we assess the performance of the proposed objectives on object-discovery tasks. Then we evaluate the efficacy of the learned representation on two distinct downstream RL tasks.

**Object Discovery Approaches**    We consider four unsupervised object discovery approaches as our base approaches: (1) Slot Attention (Locatello et al., 2020) - Slot-attention uses a top-down iterative attention mechanism to discover slots from image features ; (2) SLATE (Singh et al., 2022) - SLATE also uses slot-attention to discover slots but replaces the convolutional encoder in slot attention by a dVAE (van den Oord et al., 2017; Ramesh et al., 2021) and performs reconstruction in vector-quantzed space using an auto-regressive transformer; (2) Dinosaur (Seitzer et al., 2023) - Dinosaur applies slot attention on the features from a pretrained encoder; (4) MoTok (Bao et al., 2023) - MoTok applies slot-attention to discover objects in videos. It uses motion segmentation annotations to supervise the attention maps in slot attention. We incorporate the proposed CYCLIC objectives in each of these models.

**Datasets and Environments**    For object-discovery we consider both synthetic and real-world datasets. For the synthetic datasets, we use Shapestacks (Groth et al., 2018), ObjectsRoom (Kabra et al., 2019), ClevrTex (Karazija et al., 2021). We evaluate the segmentation performance on these synthetic datasets using the Adjusted Rand Index (ARI) (Hubert & Arabie, 1985) and reconstruction performance using the mean-squared error (MSE). Specifically, we calculate FG-ARI for all these datasets which is same as ARI but ignores the background information. For real-world datasets, we consider the task of multi-object segmentation in COCO (Lin et al., 2014) and scannet (Dai et al., 2017) datasets. For this task, we report the AP score (Everingham et al., 2014), precision score and recall score.

We also apply the proposed approach to object discovery in videos where we consider the Movi-E video dataset (Greff et al., 2022). We use MoTok (Bao et al., 2023) as our base model for this experiment. For the experiments with Dinosaur (Seitzer et al., 2023), we use the Movi-C and Movi-E datasets (Greff et al., 2022). We consider them as image datasets for these experiments rather than as video datasets.

For our downstream RL tasks we use the atari and causal world environments (Ahmed et al., 2020). In atari, we consider various different games and we report the mean returns across 10 episodes similar to (Chen et al., 2021). In causal world, we consider 2 variants of the object goal task where the agent is tasked with moving the robotic arm towards a target object. We use success rate as the performance metric in causal world.

### 4.1    OBJECT DISCOVERY

**Synthetic Datasets**    We study the object discovery performance of the proposed approach on synthetic tasks. We follow the setup used in (Dittadi et al., 2022). We augment the slot-attention auto-encoder (Locatello et al., 2020) model with the cycle consistency objectives. Slot attention uses a convolutional encoder to obtain image features and a mixture-based decoder for reconstruction (Watters et al., 2019). The encoder outputs features $\tilde{F} \in \mathbb{R}^{t \times t \times D}$, where $t = 64$. Before applying the cycle consistency objectives, we downsample the features to $\tilde{t} = 16$ and then project them using an MLP to have the same dimension as the slots. We also normalize the slots using their L2 norm before applying the cycle consistency objectives. More details regarding the architecture and hyper-parameters can be found in the Appendix.

Table 1: **Synthetic Datasets Segmentation**. In this table we compare the slot attention model augmented with the proposed CYCLIC objectives against the original slot attention model (Locatello et al., 2020). As shown in the table, we observe that the proposed objectives result in performance gains across all the considered datasets. Results averaged across 3 seeds.

| | | | ObjectsRoom | | ShapeStacks | | ClevrTex | |
|---|---|---|---|---|---|---|---|---|
| Model | $\lambda_{sfs'}$ | $\lambda_{fsf'}$ | MSE $\downarrow$ | FG-ARI $\uparrow$ | MSE $\downarrow$ | FG-ARI $\uparrow$ | MSE $\downarrow$ | FG-ARI $\uparrow$ |
| Slot-Attention | 0 | 0 | $0.0018_{\pm 0.0004}$ | $0.7819_{\pm 0.08}$ | $0.004_{\pm 0.0004}$ | $0.7738_{\pm 0.05}$ | $0.007_{\pm 0.001}$ | $0.6240_{\pm 0.223}$ [1] |
| + CYCLIC | $> 0$ | 0 | $0.0019_{\pm 0.0003}$ | $0.7832_{\pm 0.05}$ | $0.004_{\pm 0.0010}$ | $0.5491_{\pm 0.4521}$ | $0.007_{\pm 0.0001}$ | $0.6640_{\pm 0.05}$ |
| + CYCLIC | 0 | $> 0$ | $0.0015_{\pm 0.0002}$ | $0.8120_{\pm 0.06}$ | $0.004_{\pm 0.0003}$ | $0.7755_{\pm 0.06}$ | $0.007_{\pm 0.0001}$ | $0.4974_{\pm 0.03}$ |
| + CYCLIC | $> 0$ | $> 0$ | $0.0015_{\pm 0.0002}$ | $0.8169_{\pm 0.03}$ | $0.0037_{\pm 0.0001}$ | $0.7838_{\pm 0.02}$ | $0.007_{\pm 0.0001}$ | $0.7245_{\pm 0.01}$ |

---

[1] We took the result for SA from (Karazija et al., 2021), as we were unable to reproduce the same result in a statistically consistent manner. With our implementation, we got FG-ARI score of $0.5864_{\pm 0.01}$. To be fair, we have reported the score from the original paper (Karazija et al., 2021).

**Results.** Table 1 presents the quantitative results on all three datasets. We observe that augmenting Slot Attention with the proposed objectives leads to better factorization (as measured by FG-ARI) and superior reconstruction (measured by MSE). We also observe that while the presence of only one of the objectives does not affect the reconstruction performance significantly, having both objectives is crucial for achieving good factorization.

**ClevrTex Generalization.** The ClevrTex dataset provides two generalization splits which allow us to probe the generalization capabilities of the proposed approach: (1) CAMO - Contains scenes where certain objects are camouflaged, and (2) OOD - uses 25 new materials that were not seen during training. We train the models on the full training set of ClevrTex and transfer them to both the splits. Figure 3 presents the results, which demonstrate that the proposed method consistently outperforms the baseline on the generalization splits.

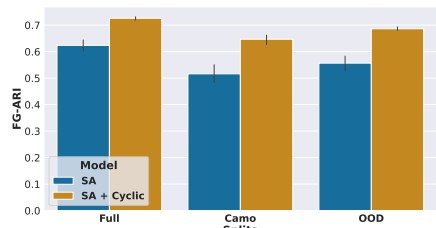

Figure 3: **ClevrTex Generalization**. Here we present transfer results on clevrtex. We find that the proposed approach outperforms the baseline on both the transfer splits. Results averaged across 5 seeds.

Table 2: **Real World Datasets Segmentation**. Here we present results for multi-object segmentation on real-world datasets. We augment the improved Slate model presented in (Jia et al., 2022) with the proposed cyclic objectives. We can see that the proposed approach outperforms all baselines on all metrics on both datasets. Results averaged across 3 seeds.

| Model | COCO | | | | Scannet | | | |
|---|---|---|---|---|---|---|---|---|
| | AP@05 ↑ | PQ ↑ | Precision ↑ | Recall ↑ | AP@05 ↑ | PQ ↑ | Precision ↑ | Recall ↑ |
| MONet (Burgess et al., 2019) | $11.8_{\pm 2.0}$ | $12.5_{\pm 1.1}$ | $16.1_{\pm 0.9}$ | $21.9_{\pm 1.7}$ | $24.8_{\pm 1.6}$ | $24.6_{\pm 1.6}$ | $31.0_{\pm 1.6}$ | $40.7_{\pm 1.8}$ |
| IODINE (Greff et al., 2019) | $4.0_{\pm 1.2}$ | $6.3_{\pm 1.2}$ | $9.9_{\pm 1.4}$ | $10.8_{\pm 2.0}$ | $10.1_{\pm 2.9}$ | $13.7_{\pm 2.7}$ | $18.6_{\pm 4.2}$ | $24.4_{\pm 3.8}$ |
| Slot Attention (Locatello et al., 2020) | $0.8_{\pm 0.3}$ | $3.5_{\pm 1.2}$ | $5.3_{\pm 1.7}$ | $7.3_{\pm 2.2}$ | $5.7_{\pm 0.3}$ | $9.0_{\pm 1.5}$ | $12.4_{\pm 2.5}$ | $18.3_{\pm 2.7}$ |
| Implicit Slot Attention (Chang et al., 2023) | $12.8_{\pm 4.8}$ | $13.7_{\pm 4.5}$ | $20.4_{\pm 6.0}$ | $24.6_{\pm 7.3}$ | $21.4_{\pm 6.8}$ | $23.4_{\pm 1.5}$ | $29.1_{\pm 7.8}$ | $34.5_{\pm 7.0}$ |
| BO-Slate (Jia et al., 2022) | $16.64_{\pm 1.00}$ | $17.48_{\pm 0.9}$ | $25.49_{\pm 1.2}$ | $31.13_{\pm 1.5}$ | $24.67_{\pm 3.2}$ | $23.55_{\pm 0.3}$ | $34.03_{\pm 0.4}$ | $38.74_{\pm 0.6}$ |
| BO-Slate + CYCLIC | $18.96_{\pm 0.9}$ | $18.81_{\pm 0.8}$ | $27.50_{\pm 1.4}$ | $33.20_{\pm 1.6}$ | $29.20_{\pm 1.1}$ | $26.09_{\pm 1.4}$ | $37.03_{\pm 1.4}$ | $42.09_{\pm 1.8}$ |

| Model | FG-ARI |
|---|---|
| MoTok | 67.38 |
| MoTok + Cyclic | 72.48 |

| Model | MOVi-C | | MOVi-E | |
|---|---|---|---|---|
| | FG-ARI | mBO | FG-ARI | mBO |
| DINOSAUR (ViT-B/8) | $68.9_{\pm 0.4}$ | $38.0_{\pm 0.2}$ | $65.1_{\pm 1.2}$ | $33.5_{\pm 0.1}$ |
| + CYCLIC | $72.4_{\pm 2.1}$ | $40.2_{\pm 0.5}$ | $69.7_{\pm 1.6}$ | $37.2_{\pm 0.4}$ |

Table 3: **Cycle Consistency Objectives with motion guidance**. Here we incorporate the cycle consistency objectives into the MoTok model presented in Bao et al. (2023). We find that incorporating the proposed objectives into MoTok results in improved performance on the considered video-based object discovery task. We consider the Movi-E dataset for this experiment.

Table 4: **Cycle Consistency Objectives with Pretrained Encoders** In this table we demonstrate the improvements achieved by the cyclic objectives when added to the DINOSAUR model from (Seitzer et al., 2023) based on the pretrained ViT-B/8 encoder. We can see that proposed method achieves superior performance on both the datasets.

**Cycle Consistency Objective for Video Datasets** We use the Movi-E video dataset for this study considering the MoTok model (Bao et al., 2023) as the base model. MoTok uses motion segmentation annotations as auxiliary information to discover objects in the video. We incorporate the cycle consistency objectives in the slot attention module from the MoTok model presented in Bao et al. (2023). We present the results for this experiment in Table 3. We find that augmenting MoTok with the cycle consistency objectives result in improved performance on the Movi-E dataset compared to the MoTok baseline.

**Cycle Consistency Objective with Pretrained Encoders** We use the DINOSAUR model (Seitzer et al., 2023) as the base model. DINOSAUR applies slot attention to the pretrained features obtained from a DINO encoder (Caron et al., 2021). For the proposed method, we add the cyclic objectives to the DINOSAUR model, applying them only to the slots obtained from the last iteration of slot attention. In this case, we do not use an EMA encoder, as the DINO image encoder is kept frozen through training. We perform this comparison on the MOVI datasets Greff et al. (2022). The results in Table 4 show that DINOSAUR augmented with cyclic objectives outperform the base DINOSAUR model on both the datasets. This further shows that the proposed cycle consistency

objectives are agnostic to the underlying object discovery approach and only require the underlying approach to use slot attention.

**Effect of Loss coefficients**   We study the effect of the loss coefficients ($\lambda_{sfs'}$ and $\lambda_{fsf'}$) on the ClevrTex dataset in Figure 4. We can see that for the variation in $\lambda_{sfs'}$ (Figure 4 (a)), the performance degrades rapidly for higher values. We conjecture that the reason behind this is that SLOT-FEATURE-SLOT consistency can be trivially satisfied if all features are mapped to one slot. Therefore, having a high $\lambda_{sfs'}$ may bias the model towards the trivial solution thereby hurting performance. From Figure 4(b), we can see that the performance variation is much more stable in the case where we vary the value of $\lambda_{fsf'}$. This shows that the model is fairly agnostic to the value of $\lambda_{fsf'}$.

**Real-World Datasets**   For these experiments, we use the BO-SLATE model (Jia et al., 2022) as our base model. BO-Slate is an improved version of Slate (Singh et al., 2022) where the main improvements come from having learnable slot initializations. For the proposed method, we add the cycle consistency objectives to the BO-SLATE model. We refer to this model as BO-Slate + Cyclic. We defer further details to the Appendix. The underlying task which we consider for real-world datasets is multi-object segmentations. The results for this task are presented in Table 2. We use the same versions of the datasets used in (Jia et al., 2022). We compare

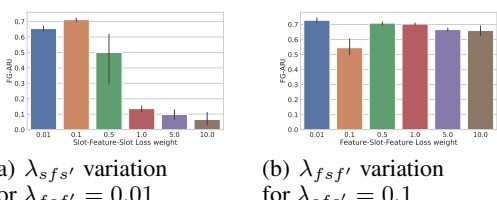

(a) $\lambda_{sfs'}$ variation for $\lambda_{fsf'} = 0.01$

(b) $\lambda_{fsf'}$ variation for $\lambda_{sfs'} = 0.1$

Figure 4: **Effect of Loss Coefficients** (a) In this we vary the SLOT-FEATURE-SLOT weight ($\lambda_{sfs'}$) and keep the FEATURE-SLOT-FEATURE weight ($\lambda_{fsf'}$) fixed to 0.01. We can see that the model reaches peak performance for $\lambda_{sfs'} = 0.1$ and the performance degrades rapidly for higher values. (b) In this we vary $\lambda_{fsf'}$ and keep $\lambda_{sfs'}$ fixed to 0.1. We can see that the performance is much more stable here as compared to varying $\lambda_{sfs'}$.

the proposed approach to various object-centric models that exist in literature. We observe that the BO-Slate model augmented with the proposed objectives outperforms all the baselines on all the metrics. More information about these metrics is presented in the appendix.

## 4.2 REPRESENTATION LEARNING FOR DOWNSTREAM RL TASKS

**Atari**   One important aspect of any representation learning method is that the learned representations should be useful in downstream tasks. In this section, we explore the usefulness of the proposed approach in the context of atari games. (Chen et al., 2021) introduced the decision transformer model which learns to play various games in the Atari suite by imitating suboptimal trajectories from a learned agent. We present more details of the decision transformer model in Appendix Section 9. In this work, we change the monolithic state representation used in decision transformer to an object-centric one.

The monolithic state representation of an observation is a $D$-dimensional vector obtained by passing the atari observations through a convolutional encoder. Note that each observation is a stack of 4 frames. To obtain the corresponding object-centric version of this, we use the convolutional encoder and the slot attention module from (Locatello et al., 2020) to encode each observation. Therefore, each observation is encoded into $N$ slots instead of a single vector.

Table 5: **Atari**. Here we present results on various games from the Atari suite. Results averaged across 5 seeds.

| Game | DT | DT + SA | DT + SA + Cyclic |
|---|---|---|---|
| Pong | $11.0_{\pm 5.727}$ | $7.4_{\pm 6.184}$ | $14.8_{\pm 2.482}$ |
| Breakout | $70.6_{\pm 20.539}$ | $93.4_{\pm 24.121}$ | $110.2_{\pm 11.107}$ |
| Seaquest | $1172.4_{\pm 175.779}$ | $444.0_{\pm 179.738}$ | $663.2_{\pm 111.014}$ |
| Qbert | $5485.2_{\pm 1995.256}$ | $5275.2_{\pm 862.894}$ | $7393.8_{\pm 1982.989}$ |
| Asterix | $523.333_{\pm 61.146}$ | $471.667_{\pm 253.388}$ | $785.0_{\pm 153.677}$ |
| Assault | $387.333_{\pm 23.099}$ | $430.667_{\pm 83.003}$ | $462.0_{\pm 128.693}$ |
| Boxing | $78.0_{\pm 1.633}$ | $77.333_{\pm 1.247}$ | $78.667_{\pm 0.943}$ |
| Carnival | $486.0_{\pm 343.872}$ | $814.0_{\pm 49.423}$ | $836.667_{\pm 91.277}$ |
| Freeway | $26.667_{\pm 0.471}$ | $21.0_{\pm 0.816}$ | $23.0_{\pm 0.816}$ |
| Crazy Climber | $76564.0_{\pm 24713.859}$ | $51490.0_{\pm 28676.178}$ | $94254.0_{\pm 7569.641}$ |
| BankHeist | $11.4_{\pm 6.974}$ | $105.0_{\pm 88.808}$ | $144.8_{\pm 116.68}$ |
| Space Invaders | $602.2_{\pm 67.972}$ | $392.0_{\pm 189.67}$ | $598.2_{\pm 52.147}$ |
| MsPacman | $1461.4_{\pm 329.76}$ | $1220.8_{\pm 237.301}$ | $1900.0_{\pm 206.364}$ |

To ensure that slots learn the correct object-centric representation we augment the decision transformer loss with the slot attention loss- $\mathcal{L} = \mathcal{L}_{DT} + \mathcal{L}_{Reconstruction}$. Additionally, we also add the

cycle consistency objectives to the loss - $\mathcal{L} = \mathcal{L}_{DT} + \mathcal{L}_{Reconstruction} + \lambda_{sfs'}\mathcal{L}_{sfs'} + \lambda_{fsf'}\mathcal{L}_{fsf'}$. We compare our method to the baseline decision transformer and an object-centric variant of decision transformer (DT + SA) where we have the slot attention style reconstruction loss but omit the cycle consistency objectives.

The performance comparison in Table 5 reveals that the decision transformer, when augmented with object-centric representations solely obtained from slot attention (DT + SA) exhibits competitive performance across most games to the original decision transformer (DT). However, when the object-centric decision transformer is combined with the proposed cycle consistency objectives, it surpasses the baseline decision transformer in 10 out of 13 games. This outcome showcases the significance of the proposed cycle consistency objective in learning strong object-centric representations capable of performing downstream tasks.

**Causal World**   We consider the object goal task from the causal world environment (Ahmed et al., 2020). We follow the same setup as (Yoon et al., 2023). In the Object Goal task, the agent controls a tri-finger robot placed in a bowl. The bowl contains a target object and a few distractor objects. The task for the agent is to move the trifinger robot towards the target objects without touching the distractors.

Similar to (Yoon et al., 2023), we first pretrain a Slate model augmented with the cycle consistency objectives on random rollouts from the object goal task. The baseline for this task is a Slate model pretrained on the same dataset of random rollouts. We visualize the learned object masks in Figure 2. We can see that the the proposed objectives enable the model to learn more accurate object masks as compared to the baseline.

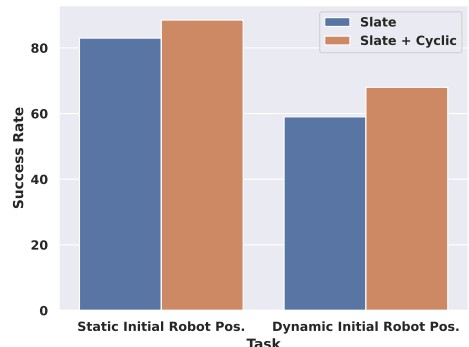

Next, we train a policy using proximal policy optimization (PPO) (Schulman et al., 2017). We use a transformer-based policy network. The inputs are the slot representations from the object-centric models along with a CLS token. The resulting output corresponding to the CLS token is used to output the action distribution and the value. The object-centric model is kept frozen at this stage.

Figure 5: **Causal World RL** In this figure, we present results for both causal world tasks. We can see that Slate models pretrained with the cyclic objectives achieve a superior success rate compared to the baseline Slate Model. Results averaged across 5 seeds.

We consider two variants of the object-goal task - (1) Static Initial Robot Position - Initial position of the tri-finger robot is the same across episodes; (2) Dynamic Initial Robot Position - Initial position of the tri-finger is different across episodes. We show results for this task in Figure 5. We can see that in both cases, models pretrained with the cyclic objective outperform those without it. We present more details about the pretraining procedure and the policy in the Appendix Section 10.

## 5   FUTURE WORK AND LIMITATIONS

Significant research efforts have been dedicated to learning object-centric representations. However, the evaluation of these representations has primarily focused on unsupervised segmentation performance using metrics like ARI or IoU. Regrettably, there is a dearth of studies demonstrating the practical utility of object-centric representations across diverse downstream tasks. To address this gap, our work takes a step forward by showcasing the effectiveness of our approach in the context of Atari and Causal World. We do not address all limitations of object-centric methods on downstream tasks, more work is still needed to showcase the effectiveness of these models on more complex and real-world downstream tasks. Moving forward, our objective is to shift our focus towards developing object-centric representations that prove valuable in a wide array of downstream tasks, spanning reinforcement learning to visual tasks like visual question answering and captioning. We aim to explore the efficacy of cycle consistency objectives in learning such representations and study what is lacking in building more pervasive object-centric representations.

## 6 ACKNOWLEDGEMENTS

This research was made possible in part by compute resources, software, and technical support provided by Mila (mila.quebec). The authors extend their appreciation to Nanda Harishankar Krishna for assistance in refining Figure 1. Gratitude is also owed to Ayush Chakravarthy and Vedant Shah for their valuable contributions during brainstorming sessions. Additionally, the authors would like to thank Nanda Harishankar Krishna, Vedant Shah, and Jithendaraa Subramanian for their thoughtful reviews and suggestions to improve this manuscript.

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

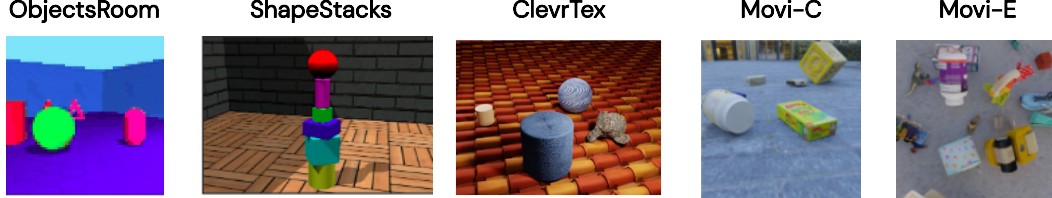

Figure 6: Here we show an example image from each synthetic dataset that we consider.

| Layer | ObjectsRoom | | | | ShapeStacks | | | | ClevrTex | | | |
|---|---|---|---|---|---|---|---|---|---|---|---|---|
| | Channels | Kernel Size | Padding | Stride | Channels | Kernel Size | Padding | Stride | Channels | Kernel Size | Padding | Stride |
| Convolutional Encoder | | | | | | | | | | | | |
| Conv | 32 | 5 | 2 | 1 | 32 | 5 | 2 | 1 | 64 | 5 | 2 | 1 |
| Conv | 32 | 5 | 2 | 1 | 32 | 5 | 2 | 1 | 64 | 5 | 2 | 1 |
| Conv | 32 | 5 | 2 | 1 | 32 | 5 | 2 | 1 | 64 | 5 | 2 | 1 |
| Conv | 32 | 5 | 2 | 1 | 32 | 5 | 2 | 1 | 64 | 5 | 2 | 1 |
| Convolutional Decoder | | | | | | | | | | | | |
| Conv. Trans | 32 | 5 | 2 | 1 | 32 | 5 | 2 | 1 | 64 | 5 | 2 | 2 |
| Conv. Trans | 32 | 5 | 2 | 1 | 32 | 5 | 2 | 1 | 64 | 5 | 2 | 2 |
| Conv. Trans | 32 | 5 | 2 | 1 | 32 | 5 | 2 | 1 | 64 | 5 | 2 | 2 |
| Conv. Trans | 4 | 5 | 1 | 1 | 32 | 5 | 2 | 1 | 4 | 5 | 2 | 1 |

Table 6: Detailed architecture for the encoder and decoder used by the slot attention model used in the synthetic dataset experiments. Note that we use relu activations after every layer except the last layer.

# APPENDIX

## 7  SYNTHETIC DATASET EXPERIMENTS

We consider the Shapestacks (Groth et al., 2018), ObjectsRoom(Kabra et al., 2019), ClevrTex (Karazija et al., 2021), and MOVi datasets (Ghorbani et al., 2020). Figure 6 shows an example image from each dataset.

**Slot Attention Implementation Details**   For ObjectsRoom, ShapeStacks, and ClevrTex, we use slot attention as our base model. Table 6 shows the detailed architecture of the convolutional encoder and decoder used by the slot attention model. Table 7 indicates values of various hyperparameters used in these experiments. For each experiment, we use 1 RTX8000 GPU.

**Ablation on $\mathcal{L}_{fsf'}$ application**   In our case, we calculate $\mathcal{L}fsf'$ as follows: $\mathcal{L}fsf' = -p^{i \to j} \log(P(f_j \mid f_i)) \quad \forall \quad i = j$. Therefore, instead of computing the loss for all pairs of $i$ and $j$, we only compute it for the cases where $i$ is equal to $j$. This design choice is made because the supervision signal, $p^{i \to j}$, is a function of the pairwise feature similarity values. Obtaining accurate pairwise feature similarity values for all $i, j$ in a model trained from scratch is challenging. Hence, we limit the loss calculation to only the diagonal elements of the matrix, where $i = j$. To assess the significance of this design choice, we compare the performance of the proposed model with a model that computes $\mathcal{L}_{fsf'}$ for all $i, j$. The results of this study are presented in Table 8. Notably, computing $\mathcal{L}_{fsf'}$ solely for $i = j$ yields considerably better performance compared to computing it for all $i, j$.

**Effect of applying the objectives on all iterations of slot attention**   One implementation detail for the proposed method is that we apply the cycle consistency objectives to slots from all iterations of slot attention. We ablate on this design choice by comparing against a model where the cycle consistency objectives are only applied to the last iteration of slot-attention. We present the results in Table 9. We can see that the performance drops significantly when the cycle consistency objectives are only applied to the last iteration thus showing the importance of applying the objectives on all iterations.

| | ObjectsRoom | ShapeStacks | ClevrTex |
|---|---|---|---|
| Num. Slots | 7 | 7 | 11 |
| Num. Iter | 3 | 3 | 3 |
| Slot size | 64 | 64 | 64 |
| MLP size | 128 | 128 | 128 |
| Batch Size | 64 | 64 | 64 |
| Optimizer | Adam | Adam | Adam |
| LR | 0.0004 | 0.0004 | 0.0004 |
| Total steps | 500k | 500k | 500k |
| Warmup Steps | 10k | 10k | 5k |
| Decay Steps | 100k | 100k | 50k |
| $\lambda_{sfs'}$ | 0.1 | 0.1 | 0.1 |
| $\lambda_{fsf'}$ | 0.01 | 0.01 | 0.01 |
| $\tau$ | 0.1 | 0.1 | 0.1 |
| $\tau_{sfs'}$ | 1 | 1 | 1 |
| $\tau_{fsf'}$ | 0.01 | 0.01 | 0.01 |
| $\theta_i$ | 0.8 | 0.8 | 0.8 |
| Downsampled feature size | $16 \times 16$ | $16 \times 16$ | $32 \times 32$ |
| EMA Decay rate | 0.995 | 0.995 | 0.995 |

Table 7: This table indicates all the values for various hyperparameters used in the synthetic dataset experiments.

| | ObjectsRoom | ShapeStacks | ClevrTex |
|---|---|---|---|
| Model | FG-ARI | FG-ARI | FG-ARI |
| SA + Cyclic ($\mathcal{L}_{fsf'}$ $\forall$ $i,j$) | $0.7341_{\pm 0.07}$ | $0.7161_{\pm 0.01}$ | $0.6349_{\pm 0.11}$ |
| SA + Cyclic ($\mathcal{L}_{fsf'}$ $\forall$ $i=j$) | $0.8169_{\pm 0.03}$ | $0.7838_{\pm 0.02}$ | $0.7245_{\pm 0.01}$ |

Table 8: Here we compare the performance of the model where $\mathcal{L}_{fsf'}$ is computed for all i, j to the model which computes $\mathcal{L}_{fsf'}$ for all i = j. We can see that the latter performs better than former thus showing the importance of computing the $\mathcal{L}_{fsf'}$ only for i = j. We perform this ablation on the ObjectsRoom, ShapeStacks, and ClevrTex datasets.

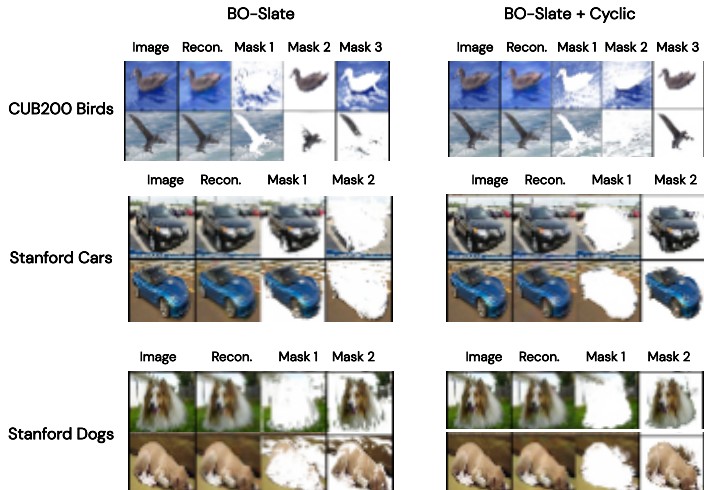

Figure 7: **Foreground Extraction Visualization**. This figure showcases the reconstruction and segmentation masks produced by the baseline model (BO-Slate) and the proposed cyclic model. Notably, we observe that the baseline model tends to mix the foreground with parts of the background, particularly for the Stanford Dogs and Stanford Cars datasets. In contrast, the use of our cyclic objectives leads to a significantly clearer separation of the foreground and background, resulting in a more accurate and refined representation of the objects of interest.

**Effect of EMA Encoder** Another implementational detail of our approach is that we use an EMA encoder to compute the feature-slot-feature supervision matrix $\tilde{\mathcal{F}}$. We examine the importance of using an EMA encoder by comparing the performance of the proposed approach on the shapestacks datasets with and without it. We present the results of this comparison in Table 10. We can see that the performance of the model with the EMA encoder is much better than the performance without it thus showing the importance of the EMA encoder.

| Model | FG-ARI |
|---|---|
| SA + Cyclic (last iteration only) | $0.5453_{\pm 0.06}$ |
| SA + Cyclic (all iterations) | $0.7245_{\pm 0.01}$ |

| Model | FG-ARI |
|---|---|
| SA + Cyclic (No EMA Encoder) | $0.7028_{\pm 0.03}$ |
| SA + Cyclic (EMA Encoder) | $0.7838_{\pm 0.02}$ |

Table 9: **Iteration Ablation** We observe that the performance suffers a significant drop when the cycle consistency objectives are only applied to the last iteration of slot attention. We run this experiment on the ClevrTex dataset.

Table 10: **EMA Encoder Ablation** We can see that the performance of the model without the EMA encoder is much worse than with it thus showing the importance of the EMA encoder. We run this experiment on the shapestacks dataset.

**Dinosaur Implementation Details** We use DINOSAUR (Seitzer et al., 2023) as the base model to which we augment the cycle consistency objectives for our experiments on the MOVi-E and MOVi-C datasets. We use a pretrained ViT-B/8 model pretrained using the approach presented in DINO (Caron et al., 2021). We use 10 slots for experiments on the MOVi-C dataset and 23 slots for experiments on the MOVi-E dataset. For both datasets, we use 3 slot attention iterations. In this case, we apply the cycle consistency objectives on the slots obtained from the last iteration of slot attention. We set $\lambda_{sfs'}$ to 5 and $\lambda_{fsf'}$ to 1. We use Adam optimizer with a learning rate of 4e-4. We run each experiment on 1 RTX8000 GPU.

# 8 REAL WORLD DATASET EXPERIMENTS

For these experiments, we use the BO-Slate (Jia et al., 2022) as the base model to which we augment the cycle consistency objectives. The values of all hyperparameters used for these experiments are shown in Table 11.

| | | |
|---|---|---|
| Training | batch size | 64 |
| | warmup steps | 10000 |
| | learning rate | 1e-4 |
| | max steps | 250k |
| | Input image size | $96 \times 96$ |
| dVAE | vocabulary size | 1024 |
| | Gumbel-Softmax annealing range | 1.0 to 0.1 |
| | Gumbel-Softmax annealing steps | 30000 |
| | lr-dVAE (no warmup) | 3e-4 |
| Transformer Decoder | layer | 4 |
| | heads | 4 |
| | dropout | 0.1 |
| | hidden dimension | 256 |
| Slot Attention Module | slot dimension | 256 |
| | iterations | 3 |
| | $\sigma$ annealing steps | 30000 |
| Cycle Consistency Objective | $\lambda_{sfs'}$ | 10 |
| | $\lambda_{fsf'}$ | 1 |
| | $\tau$ | 0.1 |
| | $\tau_{sfs'}$ | 1 |
| | $\tau_{fsf'}$ | 0.01 |
| | $\theta_i$ | 0.8 |
| | EMA Decay rate | 0.900 |
| | Downsampled Feature size | $24 \times 24$ |

Table 11: This table shows various hyperparameters used in the real-world dataset experiments where we use BO-Slate as the base model.

**Unsupervised Foreground Extraction** For foreground extraction, we use the Stanford dogs dataset (Khosla et al., 2012), Stanford cars dataset (Krause et al., 2013), CUB200 Birds dataset (Wah et al., 2011), and flowers dataset (Nilsback & Zisserman, 2006). We evaluate the performance using the IoU (Intersection over union) and the Dice metrics. IoU is calculated by dividing the overlapping area between the ground-truth and predicted masks by the union area of the ground-truth and predicted masks. Dice is calculated as twice the area of overlap between the ground-truth and predicted mask divide by the combined number of pixels between the ground-truth and predicted masks.

We use the BO-SLATE model (Jia et al., 2022) as our base model. BO-SLATE is an improved version of Slate (Singh et al., 2022) where the main improvements come from having learnable slot initializations. We apply the cycle consistency objectives as auxillary objectives to BO-Slate.

The results for this task are presented in Table 12. We observe that the proposed objectives helps the model achieve a superior performance compared to the baseline on all datasets. Additionally, in Figure 7, we visualize the reconstruction and segmentation masks from the model. We note that in certain cases, the baseline model tends to mix foreground and background information, whereas the same model augmented with the cyclic objectives is able to segregate the foreground and background information perfectly.

Table 12: **Unsupervised Foreground Extraction**. Here we present results for unsupervised foreground extraction on the Stanford Dogs, Stanford Cars, and CUB 200 birds dataset. We augment our cyclic objectives to the improved version of Slate (Singh et al., 2022) presented in (Jia et al., 2022). We can see that the performance of the Slate model improves when augmented with the proposed objectives thus showing the efficacy of our approach. Results averaged across 3 seeds.

| | Dogs | | Cars | | Birds | | Flowers | |
|---|---|---|---|---|---|---|---|---|
| Model | IoU $\uparrow$ | Dice $\uparrow$ | IoU $\uparrow$ | Dice $\uparrow$ | IoU $\uparrow$ | Dice $\uparrow$ | IoU $\uparrow$ | Dice $\uparrow$ |
| BO-Slate | $0.7875_{\pm 0.05}$ | $0.6681_{\pm 0.06}$ | $0.7686_{\pm 0.10}$ | $0.8647_{\pm 0.07}$ | $0.6129_{\pm 0.05}$ | $0.7473_{\pm 0.05}$ | $0.7461_{\pm 0.03}$ | $0.8340_{\pm 0.02}$ |
| + CYCLIC | $0.8456_{\pm 0.04}$ | $0.7462_{\pm 0.06}$ | $0.8447_{\pm 0.02}$ | $0.9145_{\pm 0.02}$ | $0.6497_{\pm 0.01}$ | $0.7797_{\pm 0.009}$ | $0.7745_{\pm 0.01}$ | $0.8525_{\pm 0.01}$ |

We follow the hyperparameters mentioned in Table 11. We use 2 slots for all foreground extraction experiments except for Birds for which we use 3 slots. To downsample the features obtained from the encoder for computing the cycle consistency objectives, we use a 2 layered convolutional network in which each layer has kernel size 4, stride 2, and padding 1. We use a relu activation between the two layers. We run each experiment on 1 RTX8000 GPU.

**Multi-Object Segmentation**   For multi-object segmentation, we use the coco (Lin et al., 2014) and scannet (Dai et al., 2017) datasets. We use the following metrics to evaluate performance -

- AP@05: AP is a metric used in object detection to measure the accuracy and relevance of detection results based on precision and recall values.

- Panoptic Quality (PQ): PQ is a comprehensive metric for evaluating object segmentation that combines segmentation quality and instance-level recognition performance into a single score.

- Precision score: Precision measures the proportion of correctly predicted foreground pixels among all the pixels predicted as foreground, indicating the accuracy of the segmentation results.

- Recall Score: Recall measures the proportion of correctly predicted foreground pixels among all the ground truth foreground pixels, indicating the completeness or coverage of the segmentation results.

We use the same hyperparameters as presented in Table 11. For calculating the cycle consistency objective, we downsample the features output by the encoder using a similar convolutional network as used in the Foreground Extraction task.

## 9   OBJECT CENTRIC MODELS IN ATARI

We follow the exact setup from decision transformer (Chen et al., 2021) for this experiment. Decision Transformer models the offline RL problem as a conditional sequence modelling task. This is done by feeding into the model the states, actions, and the return-to-go $\hat{R}_c = \sum_{c'=c}^{C} r_c$, where $c$ denotes the timesteps. This results in the following trajectory representation: $\tau = (\hat{R}_1, s_1, a_1, \hat{R}_2, s_2, a_2, \hat{R}_3, s_3, a_3, \dots)$, where $a_c$ denotes the actions and $s_c$ denotes the states. At test time, the start state $s_1$ and desired return $\hat{R}_1$ is fed into the model and it autoregressively generates the rest of the trajectory.

The original state representations $s_i$ are $D$-dimensional vectors obtained by passing the atari observations through a convolutional encoder. Note that each observation is a stack of 4 frames. To obtain the corresponding object-centric version of this, we use the convolutional encoder and the slot attention module from (Locatello et al., 2020) to encode each observation. Therefore, each observation is encoded into $N$ slots resulting in a decision transformer trajectory - $\tau = (\hat{R}_1, \{s_1^1, s_1^2 \dots, s_1^N\}, a_1, \hat{R}_2, \{s_2^1, s_2^2 \dots, s_2^N\}, a_2, \hat{R}_3, \{s_3^1, s_3^2 \dots, s_3^N\}, a_3, \dots)$.

As mentioned in the main text, we augment the action-prediction loss from decision transformer with the reconstruction loss from slot attention and the proposed cycle consistency objectives.

We use a 6-layered transformer and 8 attention heads with an embedding size 128. We use a batch size of 64. We use a context length of 50 for Pong and a context length of 30 for Seaquest, Breakout, and Qbert. We keep all other hyperparameters same as mentioned in (Chen et al., 2021). For the slot attention implementation, we define the architecture of the encoder and decoder in Table 13. The values of the other hyperparameters related to the slot attention module and the cycle consistency objectives are presented in Table 14. The input image size for the atari experiments is $84 \times 84$. We downsample the features to size $21 \times 21$ to compute the cycle consistency objectives. We use a 2-layered convolutional network for this downsampling where each layer has kernel size 4, stride 2, and padding 1.

|  | Channels | Layer | Kernel Size | Padding | Stride | output padding |
|---|---|---|---|---|---|---|
| Convolutional Encoder | Conv. | 64 | 5 | 2 | 1 | |
| | Conv. | 64 | 5 | 2 | 1 | |
| | Conv. | 64 | 5 | 2 | 1 | |
| | Conv | 64 | 5 | 2 | 1 | |
| Convolutional Decoder | Conv Trans. | 64 | 7 | 0 | 2 | 0 |
| | Conv Trans. | 32 | 3 | 1 | 2 | 1 |
| | Conv Trans. | 5 (4 for frames + 1 for mask) | 3 | 1 | 2 | 1 |

Table 13: Architecture of the encoder and decoder used in the slot attention module for the decision transformer model.

| Hyperparameter | Value |
|---|---|
| slot dimension | 256 |
| iterations | 3 |
| $\lambda_{sfs'}$ | 0.1 |
| $\lambda_{fsf'}$ | 0.01 |
| $\tau$ | 0.1 |
| $\tau_{sfs'}$ | 1 |
| $\tau_{fsf'}$ | 0.01 |
| $\theta_i$ | 0.8 |
| EMA Decay rate | 0.995 |
| Downsampled Feature size | $21 \times 21$ |

Table 14: Here we present the values for the various hyperparameters used in the slot attention module for the decision transformer experiments.

## 10 OBJECT CENTRIC MODELS IN CAUSAL WORLD

We follow the same setup as (Yoon et al., 2023) for this experiment. We first petrain the object centric model on 1000000 trajectories from the object goal task of the causal world environment. The baseline object centric model in our case is Slate. We augment it with the proposed cycle consistency objectives for our model. Both the object-centric models utilize 6 slots. We train them for 200k steps.

For training the policy, we use PPO. The agent is a transformer-based model which takes the slots as input along with a CLS token and outputs a distribution over actions and a value. We train the agent for 1000000 interaction steps.

