# OpenReview forum: "Cycle Consistency Driven Object Discovery"
_ICLR.cc/2024/Conference — ICLR 2024 poster_

### Official Review · Reviewer_nRCE · 2023-10-30

**Soundness:** 3 good
**Presentation:** 3 good
**Contribution:** 2 fair
**Rating:** 8
**Confidence:** 4

**Summary:**

The authors propose a novel regularization for object-centric learning methods which
cluster features into slots, for example Slot Attention. The proposed loss terms
regularize the cycle consistency between features and slots and vice versa:
Slot-feature-slot similarities are regularized towards an identity matrix and
feature-slot-feature similarties are regularized towards feature-feature similarities.
This regularization is shown to improve the segmentation performance compared to the
original Slot Attention model and other object-centric models. Moreover, the
regularization is shown to lead to more useful features for reinforcement learning.

**Strengths:**

- The proposed regularization is conceptually sound and leads to consistent improvements
  across the considered datasets and tasks.
- The proposed method is not only evaluated for unsupervised segmentation performance,
  but the learned, object-centric representation is also tested on a downstream task
  (reinforcement learning).

**Weaknesses:**

- It is claimed that the proposed losses address that existing approaches "rely on
  architectural priors" for learning objects. In my view that's not true. Slot
  Attention can be related to soft k-means clustering; the proposed losses enforce more
  compact and better separated clusters. But what is considered as one cluster (i.e.,
  an object) is still determined by architectural biases.
- In my view it is not sufficiently motivated why training unsupervised object-centric
  models on RGB images is the best approach for improving "object-based reasoning
  capabilities":
    - Segment Anything (Kirillov et al. 2023) suggests that generalizable object
      segmentation can be learned from limited supervised data.
    - Object-centric methods based on contrastive features such as DINO features are
      very capable, e.g. Dinosaur (Seitzer et al. 2022) or CutLER (Wang et al. 2023).
      It is argued that it "limits the applicability of the method to domains that the
      pretrained encoders are trained on". The works mentioned earlier however show that
      the resulting models work well on a range of datasets, including datasets that
      were not used to train DINO (e.g., MOVi).
    - Some works show that using additional data, such as optical flow or depth, allows
      training strong object centric methods (e.g., Karazija et al. 2022). The paper
      claims that "relying on [...] optical flow and motion is not feasible since many
      datasets and scenes do not come with this information". In my experience however,
      unlabelled video data is abundant for most practical settings.
  In summary, it is not clear to me why the restriction to unsupervised, image based
  methods trained from scratch is adequate for the goal of "developing object-based
  reasoning".
- Only FG-ARI is used as a metric for quantifying segmentation performance. It has been
  pointed out several times in the literature that this metric is problematic since it
  does not take into account whether object boundaries are accurate and favours
  undersegmentation (e.g., Engelcke et al 2020, Karazija et al. 2021, Monnier et al.
  2021). More established segmentation metrics, such as mIoU or AP, should be used that
  do not share these problems.

**Update:** The authors addressed the concerns in the rebuttal. In particular, additional experiments have shown that the improvements by the proposed method are orthogonal to other approaches, such as using contrastive features or videos. I therefore update my rating of this work.

**Questions:**

- Slot Attention can be related to soft k-means clustering, as mentioned by Locatello
  et al. 2020. From this perspective the proposed regularization terms can be
  interpreted as enforcing compact, well separated clusters. In my view the paper would
  profit from discussing the proposed loss terms from this angle.
- The Dinosaur model, which is mentioned in the paper, shows improved scalability of
  object-centric learning to real world data by applying Slot Attention in the feature
  space of a large scale contrastive model (DINO). Do the proposed regularization terms
  also improve the Dinosaur model?
- I would find the result section more natural to read if object discovery came before
  the experiments on downstream tasks.

---

> ### Author Response · Authors · 2023-11-15
> **Response to reviewer nRCE**
>
> We thank the reviewer for appreciating our experimental results and their valuable feedback for our paper.
>
> **On the point about reliance on architectral priors**
>
> The reviewer points that our claim that slot-attention based approaches rely on architectural priors while the proposed losses remove this reliance is not entirely correct.
>
> We agree with the reviewer and we apologise for any confusion caused by our claim. We would like to clarify our point. Slot-attention relies on architectural priors due to the competitive attention mechanism between the slots and the features. The proposed objectives do not remove the reliance on these architectural priors but act as a further enforcing function for the slots to learn the correct object representations which might further make it easier for the attention mechanism in slot attention to learn the correct slots thus the proposed objectives have an effect that helps or aids the architectural priors rather than reducing reliance on them. We will update this point in the main paper.

---

> > ### Author Response · Authors · 2023-11-15
> > **Motivation behind training unsupervised object-discovery methods on rgb images**
> >
> > **Motivation behind training unsupervised object-discovery methods on RGB images**
> >
> > The reviewer has questioned whether the direction of training unsupervised object-discovery methods on rgb images is still a valid direction. They have presented 3 points to illustrate their concern. We would like to address each of the 3 points -
> >
> > 1. **Segment anything shows that segmentation masks can be learned from limited supervised data** - First, we would like to point that our setting is completely different than SAM. We consider a fully unsupervised approach where in we do not have access to any ground truth object masks. On the other hand, SAM utilizes supervised masks in 3 phases of its training
> >     1.  In the first phase, the sam model is first trained with already available segmentation datasets which already have annotated masks.
> >     2.  In the second phase (Assisted Manual Phase), humans assisted by SAM annotate 120k images which are then used for training. Here most of the annotation is done by humans.
> >     3.  In the third phase (Semi-Automatic phase), more training images (180k images) are collected wherein humans only annotate those parts of the image where is sam cannot confidently output a mask.
> >
> >     Therefore, in each of the 3 phases of SAM training - some form of annotation is used for training. Therefore it already utilizes a lot more annotated data than used in any of the tasks in this paper.
> >     While it is true that SAM has strong zero-shot object-discovery performance, we actually find that it struggles a bit in slightly obscure domains that it may not have seen during training. We evaluate and compare SAM against our BO-QSA + Cyclic model on the scannet dataset. We present results in the table below -
> >
> >     |Model | AP@05 | PQ | Precision | Recall |
> >     | ---- | ----- | -- | --------- | ------ |
> >     | SAM | 15.09 | **34.32** | 35.54 | 41.29 |
> >     | BO-QSA + Cyclic | **29.20 ± 1.1** | 26.09 ± 1.4 | **37.03 ± 1.4** | **42.09 ± 1.8** |
> >
> >     We find that except for PQ, BO-QSA+Cyclic outperforms SAM in all other metrics.
> >     Another drawback for SAM is that it depends on prompts. For an image without prompts, it will first generate an exhaustive grid of points on the image for prompts and then generate a segmentation mask for each point ordered by their predicted IoU. In general, this results in more than 100 masks per image which makes SAM very slow and impractical to apply to real-time scenarios such as in the downstream tasks studied in the paper. For comparison - SAM takes **2.5 sec** to process an image while BO-QSA takes **0.02 sec** to process an image.
> >
> > 2. **Object-Centric Methods based on Contrastive Features such as DINO** - We agree that approaches like DINOSAUR perform well on datasets that dino has not been trained on such as in movi datasets. However, we would like to point that movi may still be considered fairly close to the imagenet dataset which dino was trained on since movi consists of 3d scans of real-world objects placed in real-world backgrounds [1]. The main difference is that while imagenet has images with one object, movi has images with multiple objects.
> >     On the other hand, the atari and causal world domains considered in the paper are very different than the domains that the dino encoder was trained on i.e. the imagenet dataset. Hence dinosaur may not perform very well on such cases. In fact, the atari DQN replay dataset [2] that we use in the paper for the decision transformer experiments consists of gray scale atari frames therefore DINOSAUR cannot be used for this setting since DINO requires RGB images as input.
> >     Therefore, in this paper we consider an end-to-end training approach where the encoder is trained from scratch. This allows us to evaluate the model on downstream tasks that may contain varying types of observation spaces.
> >     Furthermore, we also apply the proposed cycle consistency objectives to the DINOSAUR model and find that it improves performance of the dinosaur model on movi-c and movi-e datasets -
> >
> >     | Model | MOVi-C | MOVi-C | MOVi-E | MOVi-E |
> >     | ----- |---------------|---------------|----------------|------------|
> >     | | **FG-ARI** | **mBO** | **FG-ARI** | **mBO** |
> >     | DINOSAUR (ViT-B/8)    | 68.9 ± 0.4    | 38.0 ± 0.2 | 65.1 ± 1.2     | 33.5 ± 0.1 |
> >     | DINOSAUR + Cyclic    | **72.4 ± 2.1**| **40.2 ± 0.5**| **69.7 ± 1.6** | **37.2 ± 0.4** |
> >
> >     Therefore, while the main setting of our paper is that where the model is trained from scratch, we find that the proposed cycle consistency objectives also bring improvements to the setting where pretrained encoders such as dino are used.

---

> > > ### Author Response · Authors · 2023-11-15
> > > **Motivation behind training unsupervised object-discovery methods on rgb images (contd)**
> > >
> > > 3. **Why not use abundantly available unlabbelled video data for object-discovery** - We completely agree with the reviewer and believe that utilizing utilizing large amounts of visual data such as videos or even large number of images should be the next step forward for object-centric methods. This is one of the future works that we are currently working on but it is not that straightforward -  we find that current object-discovery approaches do not scale well with data. For example, below we present results of the dinosaur model trained on 10k samples and 100k samples of the COCO dataset. We evaluate the model on the test set of coco.
> > >
> > >     | Model | ARI | mBO |
> > >     | ----- | --- | --- |
> > >     | Dinosaur @ 10k coco samples | 34.64 | 25.70 |
> > >     | Dinosaur @ 100k coco samples | 34.91 | 25.32 |
> > >
> > >     We can see that the models performs almost similar even though one has been trained with 10x lesser data. Therefore, current state-of-the-art methods do not scale well with data. Hence, more innovation is required in designing methods that will scale well with data in order to utilize abundantly available unlabelled video or image data.

---

> > > > ### Author Response · Authors · 2023-11-15
> > > > **On evaluation metrics and relation to k-means**
> > > >
> > > > **On evaluation metrics**
> > > >
> > > > The reviewer points that we only use FG-ARI for evaluating segmentation which can be flawed. We agree that FG-ARI can be flawed. However, we would like to point that we have already used the AP and mIoU metric mentioned by the reviewer.
> > > >
> > > > In table 5 in the main paper we use AP@05, PQ, Precision and Recall for evaluating segmentation performance on coco and scannet.
> > > >
> > > > In Appendix Table 11, we use mIoU and Dice to evaluate the segmentation performance on Cars, Dogs, Birds, and Flowers datasets.
> > > >
> > > > [1] Kubric: A scalable dataset generator Greff et al 2022
> > > >
> > > > [2] https://research.google/resources/datasets/dqn-replay/
> > > >
> > > >
> > > >
> > > > **Relation of the cycle consistency objectives to k-means clustering**
> > > >
> > > > We would like to thank the reviewer for bringing up this relation. We can indeed think about the propsed objectives in relation to the soft-k means clustering which slot attention performs. Locatello et al 2020 point out that slot-attention is similar to soft k-means clustering with the differences being that they utilize a dot product similarity function and learned update rule. In view of this, the proposed cycle consistency objective can be viewed as a further enforcing the clusters to be seperated by increasing the similarity of features belonging to one cluster and decreasing the similarity of features belonging to different clusters.
> > > >
> > > > We will write this point in the main paper as the reviewer suggested.
> > > >
> > > > **On the results section**
> > > >
> > > > We thank the reviewer for the suggestion. We will move the object discovery experiments above the downstream experiments in the main paper.
> > > >
> > > >
> > > > We would like to once again thank the reviewer for their valuable feedback and time. We hope that the rebuttal has addressed all their questions. If there are further questions still remaining or if there are any more experiments that the reviewer would like us to perform we would be happy to do so.

---

> > > > > ### Comment · Reviewer_nRCE · 2023-11-17
> > > > >
> > > > > Yes I agree and apologise since my original criticism of only using FG-ARI was not
> > > > > completely justified; other metrics are used in some settings. However for me the
> > > > > paper would still be strengthend if less controversial metrics would also be used for
> > > > > the evaluation on the synthetic datasets used for the majority of experiments in the
> > > > > main paper.

---

> > > > > > ### Author Response · Authors · 2023-11-17
> > > > > > **On evaluation metrics**
> > > > > >
> > > > > > We completely agree that using other segmentation metrics might strengthen the experimental section of the synthetic datasets as well. However, we used the FG-ARI metric as that is considered standard by the various other papers in the object-discovery community. Most established baselines [1, 2] and even the baselines considered in this paper [3, 4] use FG-ARI as a metric for most experiments. Therefore, we have also opted to use FG-ARI for the synthetic experiment to ensure a fair comparison to the baselines.
> > > > > >
> > > > > > [1] Object-Centric learning with Slot Attention locatello et al 2020
> > > > > > [2] Generalization and Robustness Implications in Object-Centric Learning Dittadi et al 2022
> > > > > > [3] Improving Unsupervised Object-centric Learning with Query Optimization Jia et al 2022
> > > > > > [4] Bridging the Gap to Real-World Object-Centric Learning Seitzer et al 2022

---

> > > > ### Comment · Reviewer_nRCE · 2023-11-17
> > > >
> > > > I agree that current object-discovery approaches do not scale well. However this does
> > > > not address my original point. You do not compare to methods that improve
> > > > object-discovery by considering additional information such as optical flow. Among
> > > > others, Karazija et al. 2022 and Bao et al. 2023 show promising improvements with that
> > > > approach. In your paper your argue that these methods are not relevant competitors
> > > > since "relying on [...] optical flow and motion is not feasible since many datasets and
> > > > scenes do not come with this information"—which I think is not true for many relevant
> > > > settings.
> > > >
> > > > Bao et al. (CVPR 2023), Object Discovery from Motion-Guided Tokens.
> > > >
> > > > Karazija et al. (NeurIPS 2022), Unsupervised Multi-Object Segmentation by Predicting Probable Motion Patterns.

---

> > > > > ### Author Response · Authors · 2023-11-17
> > > > > **On comparison to approaches that use optical flow and motion**
> > > > >
> > > > > We apologise for misintepreting the reviewers original point.
> > > > >
> > > > > We will run experiments which augment one of the two papers mentioned with the reviewer with the cycle consistency objective and compare the performance. We thank the reviewer for pointing the above papers.
> > > > >
> > > > > We would also like to mention that our Dinosaur + Cyclic model already outperforms Bao et al 2023 on the Movi-E dataset.
> > > > >
> > > > > Based on Bao et al 2023, their model, MoTok, achieves a performance of 63.8 FG-ARI on the MOVI-E dataset based on table 2 here - https://arxiv.org/abs/2303.15555. Our Dinosaur + Cyclic model achieves a score of 69.7 +/- 1.6 FG-ARI on the same dataset. Furthermore, just Dinosaur achieves a performance of 65.1 FG-ARI on the same dataset which is much closer to MoTok. This further shows the effectiveness of the cycle consistency objectives.
> > > > >
> > > > > We will also try to present results of MoTok and MoTok + Cyclic before the end of the rebuttal deadline. We once again apologise for misinterpreting the reviewers comment!

---

> > > > > > ### Author Response · Authors · 2023-11-20
> > > > > > **More results comparing our approach to approaches that use optical flow and motion**
> > > > > >
> > > > > > We have run more experiments comparing our approach to [1] as suggested by the reviewer. We ran the `MoTok` model presented in this paper on the Movi-E dataset. For our approach, we integrated the cycle consistency objectives into the slot attention module from MoTok. We call this model `MoTok + Cyclic`. We present results in the table below
> > > > > >
> > > > > > | Model | FG-ARI |
> > > > > > | ------- | --------- |
> > > > > > | MoTok | 67.38 |
> > > > > > |MoTok + Cyclic | 72.48 |
> > > > > >
> > > > > > We can see that the proposed approach (MoTok + Cyclic) outperforms MoTok on MoVI-E showing the effectiveness of our approach. This further highlights that the improvements from our approach are agnostic to the base model used and only require that the base model uses a slot attention module.
> > > > > >
> > > > > >
> > > > > > [1] Bao et al. (CVPR 2023), Object Discovery from Motion-Guided Tokens.

---

> > > > > > > ### Author Response · Authors · 2023-11-20
> > > > > > > **Following up with the reviewer**
> > > > > > >
> > > > > > > We would like to follow up with the reviewer to check whether we have been able to address all their concerns and whether they would like review their rating of our paper. Furthermore, we would be happy to address any more concerns that the reviewer has or run any more experiments that the reviewer thinks may strenghthen our paper.

---

> > > > > > > > ### Comment · Reviewer_nRCE · 2023-11-20
> > > > > > > >
> > > > > > > > Thank you for your updates. My major concerns are addressed and therefore I am happy to recommend the paper for acceptance.
> > > > > > > >
> > > > > > > > > the improvements from our approach are agnostic to the base model used and only require that the base model uses a slot attention module
> > > > > > > >
> > > > > > > > This is the most convincing update in my view, demonstrating this for DINOSAUR and MoToK are great additional results. It would be great to integrate the latter in the main paper as well.
> > > > > > > >
> > > > > > > > > We completely agree that using other segmentation metrics might strengthen the experimental section of the synthetic datasets as well. However, we used the FG-ARI metric as that is considered standard by the various other papers in the object-discovery community.
> > > > > > > >
> > > > > > > > I agree that FG-ARI is important for comparisons to previous works which often quantified performance only using this metric. Nevertheless I still think that *additionally* reporting less controversial metrics (as done for a subset of the experiments) would further strengthen the paper.

---

> > > > > > > > > ### Author Response · Authors · 2023-11-20
> > > > > > > > >
> > > > > > > > > We thank the reviewer for the positive feedback and for recommending our paper for acceptance.
> > > > > > > > >
> > > > > > > > > We will add the MoTok results to the main paper too.
> > > > > > > > >
> > > > > > > > > We agree with the point on having additional metrics. We will compute the other segmentation metrics such as IoU and AP for the synthetic experiments too.

---

> > > ### Comment · Reviewer_nRCE · 2023-11-17
> > >
> > > Thank you for your detailed response.
> > >
> > > 1. **Segment anything shows that segmentation masks can be learned from limited supervised data**.
> > >    I agree that SAM is trained with supervision which is different from the setting
> > >    addressed in this work. My point was that SAM suggest that it may
> > >    be possible to train a *general* segmentation model from limited supervised data that
> > >    performs well on unseen domains. I find the results on ScanNet more convincing that
> > >    show that this is not (yet) the case and unsupervised learning performs
> > >    competitively.
> > > 2. **Object-Centric Methods based on Contrastive Features such as DINO**
> > >    - In my view the improvements by combining your method with DINOSAUR are
> > >    convincing since they show that the approach in the paper is orthogonal to other
> > >    improvements such as using pretrained contrastive features. I would like to encourage
> > >    the authors to include the results in the main paper.
> > >    - In principle, I see the point that the method in this paper can be applied to
> > >    observation spaces that do not work with the DINO encoder. However I don't think the Atari
> > >    DQN Replay dataset is a good example. Grayscale images are a subspace of RGB images,
> > >    the DINO encoder could be used by simply repeating the single channel of the
> > >    grayscale images.

---

> > > > ### Author Response · Authors · 2023-11-17
> > > >
> > > > We thank the reviewer for acknowledging the rebuttal.
> > > >
> > > > We have added the dinosaur results to the main paper now in Section 4.1. We thank the reviewer for this suggestion!
> > > >
> > > > **On using grayscale images with dino**
> > > >
> > > > The reviewer is right that we can in principal use dino with grayscale images. We would like to run this experiment. However, we follow the original decision transformer setup and in the that setup the model is fed stacks of 4 frames as observations per time step. To account for this we need to change the setup such that we feed either 1 observation (the gray scale observation stacked 3 times along the channel dimension) at a time step or feed stacks of 3 observations at a time to DINO (since dino expects 3 channel inputs). Since this might be a significant change to the original decision-transformer setup, it might be difficult to get this working before the end of the rebuttal deadline since it may not work immediately with the current hyperparameter setup. But we will try to run the decision transformer setup with a dinosaur like setup. We expect that even in this case, dinosaur augmented with the proposed cycle consistency objectives should outperform baseline dinosaur.

---

> > ### Comment · Reviewer_nRCE · 2023-11-17
> >
> > In summary, I would like to thank the authors for their detailed response including
> > many additional and convincing results which I will acknowledge in my final rating of
> > the paper. Including the additional results and changes promised to the other reviewers
> > and me while respecting the page limit seems to be challenging and will most likely
> > require compromises. Therefore it would be great if the authors could submit a revision
> > during the discussion period.

---

> ### Author Response · Authors · 2023-11-17
> **Updating the paper**
>
> We thank the reviewer again for the detailed comments. The reviewers comments have definitely helped us improve our paper. The reviewer has asked to submit an updated version of the paper incorporating the changes by all reviewers. We have submitted this updated version. Below we detail all the changes we made to the paper based on the comments from all reviewers
>
> - Putting the object-discovery experiments before the downstream tasks - Reviewer nRCE suggested this. We have made this change in the paper.
> - Added a line in the introduction that our method does not remove the reliance on architectural priors rather augments architectural priors with an extra layer of reliability - This point was brought up by Reviewer nRCE. We have made the change in the paper.
> - Relation of cycle consistency objectives to soft-k means clusterin - This point was raised by Reviewer nRCE. We have added a paragraph about this in the method section (Section 2 last paragraph)
> - Adding the results on dinosaur to the main paper - This point was suggested by Reviewer nRCE. Based on their suggestion we have added the results on dinosaur to the main paper in the object-discovery experiments section (Section 4.1 last paragraph).
> - We added a line in the future work and conclusion (Section 5) stating that we do not address all limitations of object-centric methods on downstream tasks. Rather we take a step towards making object-centric methods useful in downstream tasks. This was pointed by Reviewer sbRA.
> - Added more games for the atari experiments as requested by Reviewer DLp5.
>
> We hope that the reviewer finds these changes satisfactory and would be willing to change their rating of the paper. We would like to once again thank the reviewer and let the reviewer know that we would be happy to address any more queries that the reviewer may have regarding our paper and we would be happy to perform any more experiments that the reviewer thinks are important to improve the paper.

---

> ### Author Response · Authors · 2023-11-21
> **Update Rating ?**
>
> Dear. Reviewer,
>
> "Thank you for your updates. My major concerns are addressed and therefore I am happy to recommend the paper for acceptance."
>
> Thank you for taking time to read the rebuttal, and also for recommending the paper.
>
> Since the discussion period is coming to an end, we are wondering if it's possible to update your rating ? We are also very happy to clarify any other concerns.
>
> Thank you

---

> > ### Comment · Reviewer_nRCE · 2023-11-22
> >
> > I have just updated the rating in my original review.

---

### Official Review · Reviewer_7qi5 · 2023-10-31

**Soundness:** 3 good
**Presentation:** 3 good
**Contribution:** 3 good
**Rating:** 8
**Confidence:** 4

**Summary:**

- This work identifies two shortcomings with existing object discovery methods. The first one being excessive reliance on specific architectural priors and meticulous engineering efforts. The second shortcoming is the gap in investigating the real world application of representations learned using the discovery methods.
- To mitigate the first shortcomings, authors propose an objective function based on cycle consistency that constraints features of a single or multiple instances of an object in a scene to belong to a single slot.
- To mitigate the second limitation, authors demonstrate the effectiveness of learned representation in two downstream reinforcement learning tasks.
- Authors demonstrate that these enhancements hold true consistently across both synthetic and real world datasets showcasing the effectiveness of the proposed approach.

**Strengths:**

- The paper is well written and easy to follow.
- Authors validate all the claims made in the paper through appropriate experiments
- The proposed cycle consistency objectives are very simple and effective and I can foresee such objectives being useful for other q-former architectures like [1-2].
[1] Nicolas Carion, Francisco Massa, Gabriel Synnaeve, Nicolas Usunier, Alexander Kirillov, Sergey Zagoruyko, End-to-End Object Detection with Transformers.
[2] Junnan Li Dongxu Li Silvio Savarese Steven Hoi, BLIP-2:Bootstrapping Language-Image Pre-training with Frozen Image Encoders and Large Language Models.

**Weaknesses:**

- I do not see any major drawbacks with the current work but I believe it misses a few more analysis to show the effectiveness of the cycle consistency objectiveness.
- For example, does the Slot-feature-slot consistency objective reduce the total number of required slots creating a bottleneck? Does it have an effect on the size of the feature dimension of the slots?
- Authors showed that increasing the value of $\lambda_{s f s}$ results in a trivial solution but are there any other modes of failure?

**Questions:**

- In Eq. 8 the softmax is applied using $\tilde F$ but only along the diagonal? Can the authors elaborate what happens if the full matrix is used for the loss? Isn't that a stricter case?

**Details Of Ethics Concerns:**

I do not foresee any immediate ethical concerns with this work.

---

> ### Author Response · Authors · 2023-11-16
> **Response to reviewer 7qi5**
>
> We thank the reviewer for appreciating our paper and pointing out various use-cases of the cycle consistency objectives presented in the paper.
>
> The reviewer also points that they do not see any major drawbacks with the current work.
>
> **does the Slot-feature-slot consistency objective reduce the total number of required slots**
>
> The reviewer raises the question whether the proposed cycle-consistency objectives reduce the total number of required slots by creating a bottleneck.
>
> In our qualitative evaluation, we find that this is not the case. In general, we find that the slot utilization of models trained with the cycle consistency objectives is same as that for the models trained without it.
>
> As we can see in Figure 2 and Appendix Figure 7, both the baseline and the cyclic models utilize the same number of slots while the cyclic model result in better disentanglement of objects into slots.
>
> **Effect on Feature Dimension of the slots**
>
> We thank the reviewer for pointing this out. We haven't studied the effect of the proposed cycle consistency objectives on the dimensions of the slots or the features. We will conduct this analysis and add it to the paper.
>
> **On other modes of failure**
>
> The reviewer has asked whether there exist various modes of failures for the proposed objectives.
>
> We find that the various design decisions that we have made are important for the approach to perform well. For example, in Table 2 we have shown that having both the losses is necessary to get the best performance out of the approach. In Table 3 and 4 we have shown that applying the objectives on all iterations of slot attention and using an EMA encoder is respectively important to achieve good performance.
>
> **On using the full feature similarity matrix for the feature-slot-feature loss**
>
> The reviewer has asked about using the full feature similarity matrix for the feature-slot-feature loss.
>
> First, we would like to point that the softmax in the feature similarity matrix \tilde{F} is applied across the rows of the matrix not along the diagonal. The loss is calculated only for the diagonal elements.
>
> We have conducted an experiment wherein the loss is calculated for the entire matrix instead of only the diagonal of the matrix. We performed this experiment for the ObjectsRoom, Shapesstacks, and ClevrTex datasets. We have presented the results for this experiment in Appendix Table 8. We find that applying the loss only for the diagonal elements works consistently better than applying it for the entire matrix.
>
> We would like to once again thank the reviewer for their valuable feedback and time. We hope that the rebuttal has addressed all their questions. If there are further questions still remaining or if there are any more experiments that the reviewer would like us to perform we would be happy to do so.

---

> > ### Comment · Reviewer_7qi5 · 2023-11-16
> > **Response to authors**
> >
> > I thank the reviewer for their detailed response.
> > I am satisfied with most of the responses.
> > Authors state that the cycle consistency losses enable better disentaglement of objects. Can the authors quantify that using some form semantic/instance segmentation masks and metrics? The only evidence I see of this is Fig. 2 and it is qualitative.

---

> > > ### Author Response · Authors · 2023-11-16
> > > **Regarding segmentation metrics**
> > >
> > > We thank the reviewer for acknowledging our rebuttal. The reviewer asks for quantifying the disentanglement of the objects achieved by our approach using some segmentation metrics.
> > >
> > > In table 5, we have used the AP@05, PQ, precision and recall metrics which are generally used to evaluate segmentation models [1].
> > >
> > > In Appendix Table 11, we have also evaluated the models using mIoU and DICE which are also commonly used to measure segmentation performance [1, 2].
> > >
> > > [1] - https://kharshit.github.io/blog/2019/09/20/evaluation-metrics-for-object-detection-and-segmentation
> > >
> > > [2] - https://ilmonteux.github.io/2019/05/10/segmentation-metrics.html

---

> > > > ### Comment · Reviewer_7qi5 · 2023-11-22
> > > > **Response to authors**
> > > >
> > > > I thank the reviewers for their detailed response. I am fully convinced and would retain my rating.

---

### Official Review · Reviewer_SbRA · 2023-11-02

**Soundness:** 3 good
**Presentation:** 3 good
**Contribution:** 2 fair
**Rating:** 6
**Confidence:** 2

**Summary:**

This paper tackles object discovery and introduces additional constraints to existing slot-based methods. Specifically, two cycle consistency objectives, slot-feature-slot consistency, and feature-slot-feature consistency are explored. The authors applies the learned object-centric representations to downstream reinforcement learning tasks and demonstrates the effectiveness of the proposed method.

**Strengths:**

1. The paper is well-written and easy to follow.
2. The motivation and the development of the two consistency losses are clearly conveyed.
3. Experiments are extensively conducted to evaluate the proposed method.

**Weaknesses:**

1. The authors point out one of the limitations of existing methods that a notable gap exists for the learned object-centric representations to be used in the downstream tasks. However, it does not make sense to claim the proposed method overcomes this by achieving better performance on downstream tasks. The logic here is somewhat doubtful.

2. The main contribution of this paper is the two proposed consistency losses which constrain the model to learn discriminative slots. The technical novelty is limited.

**Questions:**

How to determine the number of slots?

As shown in Fig. 2, in the bottom right, multiple semantics exist in slot 6. Why the model cannot depart them?

---

> ### Author Response · Authors · 2023-11-16
> **Response to reviewer sbRA**
>
> We thanks the reviewer for appreciating that the paper is "well-written and easy to follow".
>
> **On the claim that the proposed method addresses the gap in downstream tasks of existing object-centric methods**
>
> The reviewer says that our claimed that the proposed approach addreses the limitations of existing object-centric models in downsteam tasks is doubtful.
>
> We apologise for the confusion. We do not want to claim that the proposed approach solves all limitations of object-centric methods on downstream tasks. Rather, we want to claim the proposed approach takes a step towards addressing the limitations of existing methods on downstream tasks.
>
> We will update the paper to reflect this.
>
>
> **Question on how to determine number of slots**
>
> In all existing object-centric papers [1, 2, 3 ...] including this one, the number of slots is a hyperparameter. The actual number of slots used by the model is problem dependent. In our experiments, we use the same number of slots used by previous papers for a given experiment.
>
> [1] Object-Centric Learning with Slot Attention Locatello et al 2020
> [2] Illiterate DALL-E Learns to Compose singh et al 2021
> [3] Improving Object-centric Learning with Query Optimization Jia et al 2022
>
>
> **On multiple semantics in slot 6 for Figure 2**
>
> The reviewer has questioned as to why multiple semantics exist in slot 6 of figure 6.
>
> In general, in slot-attention based methods it is observed that one slot ends up representing the background while other slots are representing the foreground object. This is also followed in Figure 2. We can see that slot 6 in Figure 2 (for the cyclic one) represents the base of the green arm and the bowl which constitutes the background. While the rest of the slots represent parts of the foreground. For example, slot 1 represnts the actual arm which moves the objects. Slot 4 and 5 represent the golden and blue blocks respectively.
>
> We would like to once again thank the reviewer for their valuable feedback and time. We hope that the rebuttal has addressed all their questions. If there are further questions still remaining or if there are any more experiments that the reviewer would like us to perform we would be happy to do so.

---

> > ### Comment · Reviewer_SbRA · 2023-11-22
> >
> > The rebuttal addresses my questions (across the responses to all reviewers) with qualitative and quantitative results. I confirm my positive rating for this submission.

---

### Official Review · Reviewer_DLp5 · 2023-11-02

**Soundness:** 3 good
**Presentation:** 3 good
**Contribution:** 3 good
**Rating:** 5
**Confidence:** 4

**Summary:**

This paper proposes a cyclic training loss to improve the slot-attention-based, object-centric representation learning in neural networks. Specifically, it aims to make the mapping from features to slots and from slots to features more distinct. Additionally, it emphasizes applying the learned representations to downstream tasks.

**Strengths:**

It is a reasonable idea to use additional regularization terms in the training objective to make each slot in slot attention represent a more distinct concept. The paper presents this idea clearly.
Experimental results show improved performance across various downstream tasks, including four Atari games, object discovery, and COCO/Scannet segmentation.
For segmentation tasks, it is interesting to see that additional cyclic losses are helpful with BO-Slate, as BO-Slate's optimization method should already aim to enhance disentangling between slots.

**Weaknesses:**

The experimental results show improvement, the overview accuracy level is low for real-world object discovery tasks. I doubt if adding more constraints on the disentanglement of representation is a promising direction.

The results on Atari games are a bit mixed. Also, why only evaluate it on four games?

**Questions:**

As there are already some works aiming to improve the slot attention-based method, the significance of this paper would be enhanced if it could show more compelling practical results, demonstrating that the overall direction is promising.
Why are the results limited to only 4 Atari games? Do the results generalize to more games?

---

> ### Author Response · Authors · 2023-11-15
> **Response to Reviewer DLp5**
>
> We thank the reviewer for their valuable feedback and we are glad that the reviewer finds the paper clearly written.
>
>
> **Regarding whether the proposed objectives are a promising research direction**
>
> The reviewer questions whether adding objectives that constrain the representation to be disentangled is a promising research direction.
>
> Many existing unsupervised object discovery approaches that are based on slot attention use pixel-level reconstruction objecives [1, 2, 3 ...]. We have already seen in self-supervised learning literature [4, 5, 6, ...] that objectives which operate in the latent space are far superior at learning strong representations as compared to objectives that directly operate in the pixel space. Pixel-space objectives may be prone to capture unecessary details while latent-space objectives learn stronger representations that are more conducive to downstream tasks.
>
> To the best of our knowledge there are only two papers on unsupervised object-discovery that utilize objectives in the latent space [7, 8], both these papers use pretrained and frozen encoders thus the objectives dont have any effect on the encoder itself. Ours is one of the first unsupervised object-discovery works which utilizes latent space objectives and training happens end-to-end i.e. the objectives affect the encoder itself. By utilizing such an approach, we hope that the model will not only learn a disentangled representation but also learn to represent objects in a meaningful manner.
>
> The proposed objectives are inspired by the cycle consistency objectives that already exist in self-supervised learning literature for videos [9] and images [10] and have shown to learn strong self-supervised representations when trained on a lot of data. The promise of our approach comes from the fact that it opens a new direction of research on designing self-supervised learning objectives for end-to-end learning of object discovery models. In self-supervised learning literature it has been shown that models trained with self-supervised learning objectives [4, 5, 6] with a lot of data result in very strong representations. Even though it might seem that the current paper does not show very strong improvements, we hope that we will see similar scaling trends with respect to data in object-discovery methods when trained with objectives such as the ones presented in this paper.
>
> Furthermore, we would also like to point that in terms of results we do outperform the current state-of-the-art model in end-to-end object centric learning (BO-QSA [3]) on various real-world datasets - Table 5 and Appendix Table 11.
>
> In summary, we firmly believe this paper showcases promising results and paves the way for promising future works to stem from its foundations.
>
>
> [1] Object-Centric Learning with Slot Attention Locatello et al 2020
>
> [2] Illiterate DALL-E Learns to Compose singh et al 2021
>
> [3] Improving Object-centric Learning with Query Optimization Jia et al 2022
>
> [4] Emerging Properties in Self-Supervised Vision Transformers Caron et al 2021
>
> [5] Bootstrap your own latent: A new approach to self-supervised Learning Grill et al 2020
>
> [6] Barlow Twins: Self-Supervised Learning via Redundancy Reduction Zbontar et al 2021
>
> [7] Bridging the Gap to Real-World Object-Centric Learning Seitzer et al 2022
>
> [8] Object-centric Learning with Cyclic Walks between Parts and Whole Wang et al 2023
>
> [9] Space-Time Correspondence as a Contrastive Random Walk Jabri et al 2020
>
> [10] Leveraging Unpaired Data for Vision-Language Generative Models via Cycle Consistency (ICLR 2024 submission)

---

> > ### Author Response · Authors · 2023-11-15
> > **Atari Results**
> >
> > **On Atari results**
> >
> > The reviewer asked why we have evaluated on only 4 atari games.
> >
> > We followed the setting of the original decision transformer paper [1]. In the decision transformer paper, they studied only 4 games - Pong, Qbert, Seaquest, and Breakout. Therefore, we also studied the same 4 games in this work.
> >
> > Based on the reviewers suggestions we also ran a few more atari games. We present the results in the table below -
> >
> > | Game     | Model         | Returns      |
> > |----------|-------------------|------------------|
> > | Asterix  | DT                | 523.333+/-61.146 |
> > |          | DT + SA           | 471.667+/-253.388|
> > |          | DT + SA + Cyclic  | **785.0+/-153.677**  |
> > | Assault  | DT                | 387.333+/-23.099 |
> > |          | DT + SA           | 430.667+/-83.003 |
> > |          | DT + SA + Cyclic  | **462.0+/-128.693**  |
> > | Boxing   | DT                | 78.0+/-1.633     |
> > |          | DT + SA           | 77.333+/-1.247   |
> > |          | DT + SA + Cyclic  | **78.667+/-0.943**  |
> > | Carnival | DT                | 486.0+/-343.872 |
> > |          | DT + SA           | 814.0+/-49.423  |
> > |          | DT + SA + Cyclic  | **836.667+/-91.277** |
> > | Freeway  | DT                | **26.667+/-0.471**   |
> > |          | DT + SA           | 21.0+/-0.816     |
> > |          | DT + SA + Cyclic  | 23.0+/-0.816     |
> >
> > In the above table, we find that the proposed DT + Slot-attention (SA) + Cyclic approach outperforms the baselines - DT and DT + SA in all games except freeway. This further shows that the proposed approach is effective on more games than showed in the paper.
> >
> > Due to the limited time of the rebuttal we could only run 5 additional games. We will add more results on more games after the rebuttal.
> >
> > [1] Decision Transformer: Reinforcement Learning via Sequence Modeling Chen and Lu et al 2021

---

> > > ### Author Response · Authors · 2023-11-15
> > > **More practical results**
> > >
> > > **More Compelling Practical Results**
> > >
> > > The reviewer points out that since there are many papers that aim to improve upon slot-attention, this paper would be improved with more compelling practical results.
> > >
> > > The reviewer is right that there are many papers that propose various improvements of slot attention. However, as we pointed out before, our work is unique in the sense that it is the first work that introduces objectives in the latent space for object discovery for end-to-end learning of object representation. In terms of practicality of results, we compare our approach to two state-of-the-art models -
> > >
> > > 1. **BO-QSA** [1] - BO-QSA, to the best of our knowledge, is the state-of-the-art object-centric model in end-to-end learned object-centric models i.e. in models where the encoder is trained from scratch. In Table 5 we have shown that when BO-QSA is combined with the proposed cycle-consistency objectives we are able to achieve superior performance than BO-QSA. Secondly, in Appendix Table 11 we have shown BO-QSA + Cyclic outperforms BO-QSA in a range of foreground-extraction tasks from various datasets.
> > > 2. **Dinosaur** [2] - Dinosaur, to the best of our knowledge, is the state-of-the-art model for object-discovery in current literature. It relies on pretrained and frozen encoders therefore the encoder is not trained from scratch. We incorporate the proposed cycle consistency objectives into dinosaur and find that we get improved performance across the movi-c and movi-e datasets as presented below
> > >
> > > | Model | MOVi-C | MOVi-C | MOVi-E | MOVi-E |
> > > | ----- |---------------|---------------|----------------|------------|
> > > | | **FG-ARI** | **mBO** | **FG-ARI** | **mBO** |
> > > | DINOSAUR (ViT-B/8)    | 68.9 ± 0.4    | 38.0 ± 0.2 | 65.1 ± 1.2     | 33.5 ± 0.1 |
> > > | DINOSAUR + Cyclic    | **72.4 ± 2.1**| **40.2 ± 0.5**| **69.7 ± 1.6** | **37.2 ± 0.4** |
> > >
> > > Further details about the dinosaur experiment is presented in Appendix Section 6.
> > >
> > > [1] Improving Object-centric Learning with Query Optimization Jia et al 2022
> > >
> > > [2] Bridging the Gap to Real-World Object-Centric Learning Seitzer et al 2022
> > >
> > >
> > > We would like to once again thank the reviewer for their valuable feedback and time. We hope that the rebuttal has addressed all their questions. If there are further questions still remaining or if there are any more experiments that the reviewer would like us to perform we would be happy to do so.

---

> > > ### Author Response · Authors · 2023-11-17
> > > **Updating the paper**
> > >
> > > We thank the reviewer again for the detailed comments. The reviewers comments have definitely helped us improve our paper. We have updated our paper with the suggestions from all reviewers. Below, we detail all the changes we have made.
> > >
> > > - Added more games for the atari experiments as requested by Reviewer DLp5.
> > > - Putting the object-discovery experiments before the downstream tasks - Reviewer nRCE suggested this. We have made this change in the paper.
> > > - Added a line in the introduction that our method does not remove the reliance on architectural priors rather augments architectural priors with an extra layer of reliability - This point was brought up by Reviewer nRCE. We have made the change in the paper.
> > > - Relation of cycle consistency objectives to soft-k means clustering - This point was raised by Reviewer nRCE. We have added a paragraph about this in the method section (Section 2 last paragraph)
> > > - Adding the results on dinosaur to the main paper - This point was suggested by Reviewer nRCE. Based on their suggestion we have added the results on dinosaur to the main paper in the object-discovery experiments section (Section 4.1 last paragraph).
> > > - We added a line in the future work and conclusion (Section 5) stating that we do not address all limitations of object-centric methods on downstream tasks. Rather we take a step towards making object-centric methods useful in downstream tasks. This was pointed by Reviewer sbRA.
> > >
> > > We hope that the reviewer finds these changes satisfactory and would be willing to change their rating of the paper. We would like to once again thank the reviewer and let the reviewer know that we would be happy to address any more queries that the reviewer may have regarding our paper and we would be happy to perform any more experiments that the reviewer thinks are important to improve the paper.

---

> ### Author Response · Authors · 2023-11-20
> **More atari results**
>
> We were able to run our approach on 4 more atari games
>
> | Game           | Model             | Returns              |
> |----------------|-------------------|----------------------|
> | Crazy Climber  | DT                | 76564.0+/-24713.859  |
> |                | DT + SA           | 51490.0+/-28676.178  |
> |                | DT + SA + Cyclic  | **94254.0+/-7569.641**|
> | BankHeist      | DT                | 11.4+/-6.974          |
> |                | DT + SA           | 105.0+/-88.808       |
> |                | DT + SA + Cyclic  | **144.8+/-116.68**    |
> | Space Invaders | DT                | **602.2+/-67.972**       |
> |                | DT + SA           | 392.0+/-189.67       |
> |                | DT + SA + Cyclic  | 598.2+/-52.147   |
> | MsPacman       | DT                | 1461.4+/-329.76      |
> |                | DT + SA           | 1220.8+/-237.301     |
> |                | DT + SA + Cyclic  | **1900.0+/-206.364** |
>
> We can see that here also in 3/4 games DT + SA + Cyclic is the best performing model. Even in space invaders, where DT + SA + Cyclic performs worse than DT, we find that the performance of both the models is actually quite comparable.
>
> In total, we have run our approach on 13 atari games and find that it outperforms the baseline 10/13 games.

---

> > ### Author Response · Authors · 2023-11-20
> > **Following up with the reviewer**
> >
> > We would like to follow up with the reviewer to check whether we have been able to address all their concerns and whether they would like review their rating of our paper. Furthermore, we would be happy to address any more concerns that the reviewer has or run any more experiments that the reviewer thinks may strenghthen our paper.

---

> > > ### Comment · Area_Chair_gSnY · 2023-12-05
> > >
> > > Reviewer DLp5 - We are close to the end of the discussion phase. I wonder if you could review the responses from the authors and other reviewers’ comments and consider whether you would like to maintain or modify your rating. Thank you.

---

### Author Response · Authors · 2023-11-21
**General Comment**

We would like to thank the reviewers for giving us valuable feedback for our paper. As the discussion phase is drawing to a close, in this general comment, we would like to summarize the changes we have made to our paper in response to the various concerns raised by each of the reviewers.

First, we would like to mention the new experiments we have performed during the rebuttal phase -
1. We have incorporated the cycle consistency objecive in the dinosaur model (Seitzer et al 2022) and showed that it improves performance over the base dinosaur model on image based object-discivery tasks. Here we considered the Movi-c and Movi-e datastes. We have updated the paper with this result (Section 4.1 page 7). **This experiment addresses Reviewer nRCE's question on incorporating cycle consistency in dinosaur and also Reviewer DLp5's concern for more practical results**.
2. We have also incorporated the cycle consistency objectives in the MoTok model (Bao et al 2023) and showed that it improves performance over the base MoTok model. Here we considered the setting of video-based object-discovery using the Movi-e dataset. We have updated the paper with this result (Section 4.1 page 7). **This experiment addresses Reviewer nRCE's question on comparing our approach to methods that use optical flow and motion and also Reviewer DLp5's concern for more practical results**.
3. We added results on 9 additional games for the decision transformer based atari experiments in the paper. We have updated table 5 in the paper with the results. **This experiment addresses Reviewer DLp5's request for running on more atari games**.

**Reviewer DLp5 (Original Rating - 5)**

| Concern | Response | Change in paper ? |
| ------- | -------- | ----------------- |
| "I doubt if adding more constraints on the disentanglement of representation is a promising direction." | We have addressed this in the rebuttal - https://openreview.net/forum?id=f1xnBr4WD6&noteId=YAyYymJplG | - |
| We have only evaluated on 4 atari games | We evaluated the approach on 9 more atari games and have presented the results in the rebuttal. | We have updated the paper with the results for all 13 games in Table 5 |
| We should show more compelling practical results | We have added two more results that show that proposed objective is agnostic to the underlying object-discovery approach and only requires that the underlying approach uses slot-attention. **Reviewer nRCE has said that this is the "most convincing update"** - 1. We have applied the objectives to the dinosaur (Sietzer et al 2022) model and showed improvements for movi-c and movi-e. 2. We have applied the objectives to the MoTok model (Bao et al 2023) and showed improvements on the Movi-E dataset. | We have updated the paper with these results (Section 4.1 Page 7) |

These are the 3 main concerns raised by the reviewer which we have addressed. Reviewer DLp5 has not responded to our rebuttal yet but we hope that they find our responses satisfactory and would be willing to accept this paper.

We would be happy to clarify any more concerns that they might have.

---

> ### Author Response · Authors · 2023-11-21
> **General Comment (continued)**
>
> **Reviewer nRCE (Original Rating - 5, Updated rating - ?)**
>
> We thank the Reviewer nRCE for engaging with us in a discussion during the rebuttal period. Their suggestions have been very helpful and have allowed us to improve our paper. **While initially Reviewer nRCE gave our paper a rating of 5, after our rebuttal they have mentioned that - "My major concerns are addressed and therefore I am happy to recommend the paper for acceptance."** We hope that they will raise the score of the paper accordingly.
>
> Below, we summarize their concerns and our responses.
>
> | Concern | Response | Change in the paper? |
> | ------ | --------- | -------------------- |
> | Our approach does not remove reliance on architectural priors | We agree with the reviewer and have clarified our claim in https://openreview.net/forum?id=f1xnBr4WD6&noteId=0V75iY0SgI. We have also updated the paper with the clarification | We have clarified this in the introduction (marked in red) |
> | Comparison to SAM? | We have compared our approach to SAM and showed that SAM does not yet outperform unsupervised approaches https://openreview.net/forum?id=f1xnBr4WD6&noteId=RArTIu56CG | - |
> | Approaches based on DINO perform well on domains not used while training DINO | We have addressed this in https://openreview.net/forum?id=f1xnBr4WD6&noteId=RArTIu56CG. | We have updated the paper in section 4.1 page 7 with results where we incorporate the proposed objectives in dinosaur which is based on dino seitzer et al 2022. |
> | Why not use abundantly available video data or compare with approaches that utilize video and auxillary data such as optical flow or motion information? | We have addressed this in https://openreview.net/forum?id=f1xnBr4WD6&noteId=Cj51XEAYT5. Additionally we have applied our approach to an approach that uses motion information in videos for object-discovery called MoTok (Bao et al 2023). We find that MoTok + Cyclic outperforms MoTok. | We have updated the paper with the MoTok results in Section 4.1 page 7. |
> | Why not use evaluation metrics like AP and IoU? | We have addressed in https://openreview.net/forum?id=f1xnBr4WD6&noteId=k6OTh3bjlg | - |
> | Relation of the proposed objecives to k-means clustering as pointed in slot-attention paper? | We have addressed in https://openreview.net/forum?id=f1xnBr4WD6&noteId=k6OTh3bjlg | We have updated the paper with a paragraph on this relation in Section 2 last paragraph. |
> | The reviewer requested to change the results section to have the object-discovery experiments before the downstream experiments. | | We have updated the paper as suggested. |
>
> We once again would like to thank the reviewer for engaging in a discussion with us and helping us improve the paper.
>
> **Reviewer 7qi5 (Original Rating - 8)**
>
> We would like to thank the reviewer for rating our paper highly and engaging with us in a discussion regarding the paper.
>
> Below, we summarize their concerns and our responses.
> | Concern | Response | Change in Paper ? |
> | ----- | ----- | ----- |
> | Does the slot-feature-slot consistency objectives reduce the total number of required slots | We have addressed in https://openreview.net/forum?id=f1xnBr4WD6&noteId=fLLT2VCpYx | |
> | Effect of the objectives on the dimensions of the features and slots | We haven't studied this yet, we will add this study to the paper after the discussion phase | |
> | Other modes of failure of the cycle consistency objectives | We have addressed in https://openreview.net/forum?id=f1xnBr4WD6&noteId=fLLT2VCpYx | |
> | On using the full feature similarity matrix for the feature-slot-feature loss | We have addressed in https://openreview.net/forum?id=f1xnBr4WD6&noteId=fLLT2VCpYx | |
> | Using segmentation metrics to quantify disentanglement | We have addressed in https://openreview.net/forum?id=f1xnBr4WD6&noteId=xmheSBmP85 | |
>
>
> **Reviewer sbRA (Original Rating - 6)**
>
> We thank the reviewer for rating to accept our paper. We have updated our paper based on some of the suggestions from the reviewer.
>
> Below, we summarize their concerns and our responses.
>
> | Concern | Response | Change in Paper? |
> | ----- | ---- | -- |
> | Our approach does not completely address the performance gap of object-centric methods on downstream tasks | We agree with the reviewer and have clarified this in https://openreview.net/forum?id=f1xnBr4WD6&noteId=UrZiG0gucD | We have also updated the paper to clarify our claim in Section 5 marked in red. |
> | How to determine number of slots? | We have addressed in https://openreview.net/forum?id=f1xnBr4WD6&noteId=UrZiG0gucD | |
> | Why multiple semantics exist in Slot 6 in Figure 2? | We have addressed in https://openreview.net/forum?id=f1xnBr4WD6&noteId=UrZiG0gucD | |
>
> We once again thank the reviewer for their valuable feedback. We hope that our rebuttal has addressed their concerns.

---

### Meta-Review · Area_Chair_gSnY · 2023-12-12

**Metareview:**

The paper integrates cycle consistency as regularization into the slot attention mechanism, enhancing object-centric representation learning for self-supervised object discovery. This is implemented through two cycle consistency objectives: slot-feature-slot consistency and feature-slot-feature consistency, ensuring each object is linked to a unique slot. Beyond the primary object discovery task, the paper explores the application of the acquired object-centric representation in downstream reinforcement learning tasks.

The initial submission had notable shortcomings, particularly in its absence of comparisons with recently introduced, more advanced slot attention variants and its limited evaluation on reinforcement learning benchmarks. These concerns were the primary focus of the reviewers. During the author-reviewer discussion phase, the authors diligently addressed these issues, notably by incorporating the cycle consistency objectives into the DINOSAUR model (Seitzer et al.) and the MoTok model (Bao et al.) and expanding the reinforcement learning evaluation to include more Atari games. At the end of the discussion phase, two reviewers gave a score of 8 (accept, good paper), one reviewer gave a score of 6 (marginally above the acceptance threshold), and one reviewer gave a score of 5 (marginally below the acceptance threshold).

While there are remaining unsolved issues, the reviewers are mostly positive for the acceptance since the main concerns have been addressed. There is no basis to overturn reviews. The area chairs agree with this recommendation. In the camera ready, the authors should include the additional experiments and explanations presented in the discussion. Also, both the reviewers and area chairs feel that some statements in the paper are inaccurate, and it is suggested to revise them in the camera ready.

Despite some unresolved issues, the reviewers lean towards acceptance as the main concerns have been addressed. There is no compelling reason to overturn the reviews, and the area chairs concur with this recommendation. For the camera-ready version, it is advised to incorporate the additional experiments and explanations on realistic datasets presented during the discussion. Additionally, both reviewers and area chairs recommend revising certain inaccurate statements in the paper.

**Justification For Why Not Higher Score:**

The cycle consistency objectives have been widely employed in various contexts. They also demonstrate effectiveness in this paper with a relatively simple design. However, there is a lack of in-depth analysis regarding the importance and sufficiency of this simplicity. In addition, it would be more convincing to show comprehensive results on both video and realistic datasets (such as Davis or youtubevis). For downstream reinforcement learning tasks, a comparison with other slot attention variants would strengthen the argument. These limitations prevent the area chairs from recommending a higher score.

**Justification For Why Not Lower Score:**

Despite some unresolved issues, the reviewers lean towards acceptance as the main concerns have been addressed. There is no compelling reason to overturn the reviews, and the area chairs concur with this recommendation. For the camera-ready version, it is advised to incorporate the additional experiments and explanations on realistic datasets presented during the discussion. Additionally, both reviewers and area chairs recommend revising certain inaccurate statements in the paper.

---

### Decision · Program_Chairs · 2024-01-16

Accept (poster)